# Space as a scaffold for rotational generalisation of abstract concepts

Jacques Pesnot Lerousseau*, Christopher Summerfield

Department of Experimental Psychology, University of Oxford, Oxford, United Kingdom

**Abstract** Learning invariances allows us to generalise. In the visual modality, invariant representations allow us to recognise objects despite translations or rotations in physical space. However, how we learn the invariances that allow us to generalise abstract patterns of sensory data ('concepts') is a longstanding puzzle. Here, we study how humans generalise relational patterns in stimulation sequences that are defined by either transitions on a nonspatial two-dimensional feature manifold, or by transitions in physical space. We measure rotational generalisation, i.e., the ability to recognise concepts even when their corresponding transition vectors are rotated. We find that humans naturally generalise to rotated exemplars when stimuli are defined in physical space, but not when they are defined as positions on a nonspatial feature manifold. However, if participants are first pre-trained to map auditory or visual features to spatial locations, then rotational generalisation becomes possible even in nonspatial domains. These results imply that space acts as a scaffold for learning more abstract conceptual invariances.

**\*For correspondence:**
jacques.pesnot@hotmail.fr

**Competing interest:** The authors declare that no competing interests exist.

## eLife assessment

These ingenious and thoughtful studies present **important** findings concerning how people can represent and generalise abstract patterns of sensory data. The issue of generalization is a core topic in neuroscience and psychology, relevant across a wide range of areas, and the findings will be of interest to researchers across areas in perception, learning and cognitive science. The findings are **convincing** in this setting, but future research must establish their generality and interrogate the precise nature of the underlying mechanism.

## Introduction

To recognise objects and events in the natural world, humans form mental representations that are invariant to transformation. The existence of invariant representations allows entities to be recognised and categorised despite changes in their surface properties, which is called 'generalisation'. The formation of invariances has been most extensively studied in the case of visual object recognition. For example, we have no trouble recognising a teapot that is moved to a new location (translated), tipped on its side (rotated) or viewed from afar (rescaled). How we do so has provoked diverse theories based on assembly from geometric primitives (*Biederman, 1987*; *Marr, 1982*), associative learning (*Rock and DiVita, 1987*; *Wallis and Bülthoff, 1999*), and function approximation in deep networks (*Lindsay, 2021*).

A core problem in cognitive science, however, is how we form invariances over entities that are defined by more abstract relational properties. Here, we use the term 'concepts' to refer to objects or events that are defined by shared relations among features that may unfold in space, or time, or both, in any modality (*Tenenbaum et al., 2011*). For example, the concept of a 'tree' implies an entity whose structure is defined by a nested hierarchy, whether this is a physical object whose parts are

arranged in space (such as an oak tree in a forest) or a more abstract data structure (such as a family tree or taxonomic tree). The concept of a 'ring' implies an entity whose features are arranged cyclically, whether a physical ring (worn on the finger), the (circular) temporal pattern of tones in a peal of bells, or the periodicity in the passage of the seasons (*Kemp and Tenenbaum, 2008*). Despite great changes in the surface properties of oak trees, family trees, and taxonomic trees, humans perceive them as different instances of a more abstract concept defined by the same relational structure. The human ability to readily form invariances over abstract concepts remains a puzzle for both cognitive and neural scientists hoping to understand neural computations, and a challenge for AI researchers wishing to build intelligent agents.

One prominent theory argues that we learn invariant concepts because of the way the brain represents physical space (*Behrens et al., 2018*; *Bellmund et al., 2018*; *Summerfield et al., 2020*; *Gärdenfors, 2000*; *Tversky, 2001*). This argument states that neurons coding for positions in either egocentric (viewer-centred) or allocentric (world-centred) space can be recycled to represent locations in more abstract spaces, defined by continuous variation in features (e.g. red to blue, quiet to loud). This theory is backed up by proof-of-concept computational simulations (*Whittington et al., 2020*), and by findings that brain regions thought to be critical for spatial cognition in mammals (such as the hippocampal-entorhinal complex and parietal cortex) exhibit neural codes that are invariant to relational transformations of nonspatial stimuli (*Park et al., 2021*; *Park et al., 2020*; *Mack et al., 2018*; *Constantinescu et al., 2016*; *Viganò and Piazza, 2020*). However, whilst promising, this theory lacks direct empirical evidence. Here, we set out to provide a strong test of the idea that learning about physical space scaffolds conceptual generalisation. Our focus is on the ability to generalise knowledge about the relations among items in a sequence as they are translated or rotated through both spatial and nonspatial domains.

In the four studies described here, participants made category judgments about a sequence of four successive stimuli in either auditory, visual, or spatial modalities. In auditory and visual modalities, the stimulus was drawn from a two-dimensional (2D) feature manifold (e.g. a bivariate 'space' defined by colour and shape in the visual modality or pitch and timbre in the auditory modality). In the spatial modality, each stimulus was a position in physical space (e.g. an $x$ and $y$ coordinates). Concepts were defined by a common pattern of transitions through either feature space or physical space. Our research question concerned the conditions under which concepts could be recognised, even if their corresponding transition vectors had been translated or rotated. We studied generalisation of transition vectors both within the same feature space and to new feature spaces in the same modality.

Our studies measure the tendency to generalise by both translation and rotation. Conceptual translation occurs when feature values are shifted in either dimension, but with no change in their relational pattern. There is already good evidence that nonspatial concepts are represented in a translation-invariant format. For example, in the auditory domain, we can recognise 'auditory objects' that are translated in feature space (e.g. pitch and timbre). This occurs when we understand the same sentence from different speakers, or identify the same melody played with different musical instruments (*Winkler et al., 2009*; *Griffiths and Warren, 2004*). However, much less is known about the learning of rotational invariances for abstract concepts. In physical space, we readily learn rotation-invariant object representations (allowing us to recognise an upside-down teapot), and the computational mechanisms by which we do so have been a major fulcrum of debate in the vision sciences (*Rock and DiVita, 1987*; *Wallis and Bülthoff, 1999*). But whether participants can learn rotationally invariant concepts in nonspatial domains, i.e., those that are defined by sequences of visual and auditory features (rather than by locations in physical space, defined in Cartesian or polar coordinates) is not known. In the current study, we first test this, and find that naively, they cannot. Next, turning to our main hypothesis, we then ask if first teaching participants to map nonspatial features to spatial locations (providing a spatial scaffold) allows the learning of rotational invariances, even in nonspatial modalities. We find that it does. This shows that a form of generalisation that is not usually possible for humans becomes possible when their understanding of the concept is 'scaffolded' by first learning a corresponding spatial representation. This thus supports the theory that abstract concept learning is linked to our understanding of physical space.

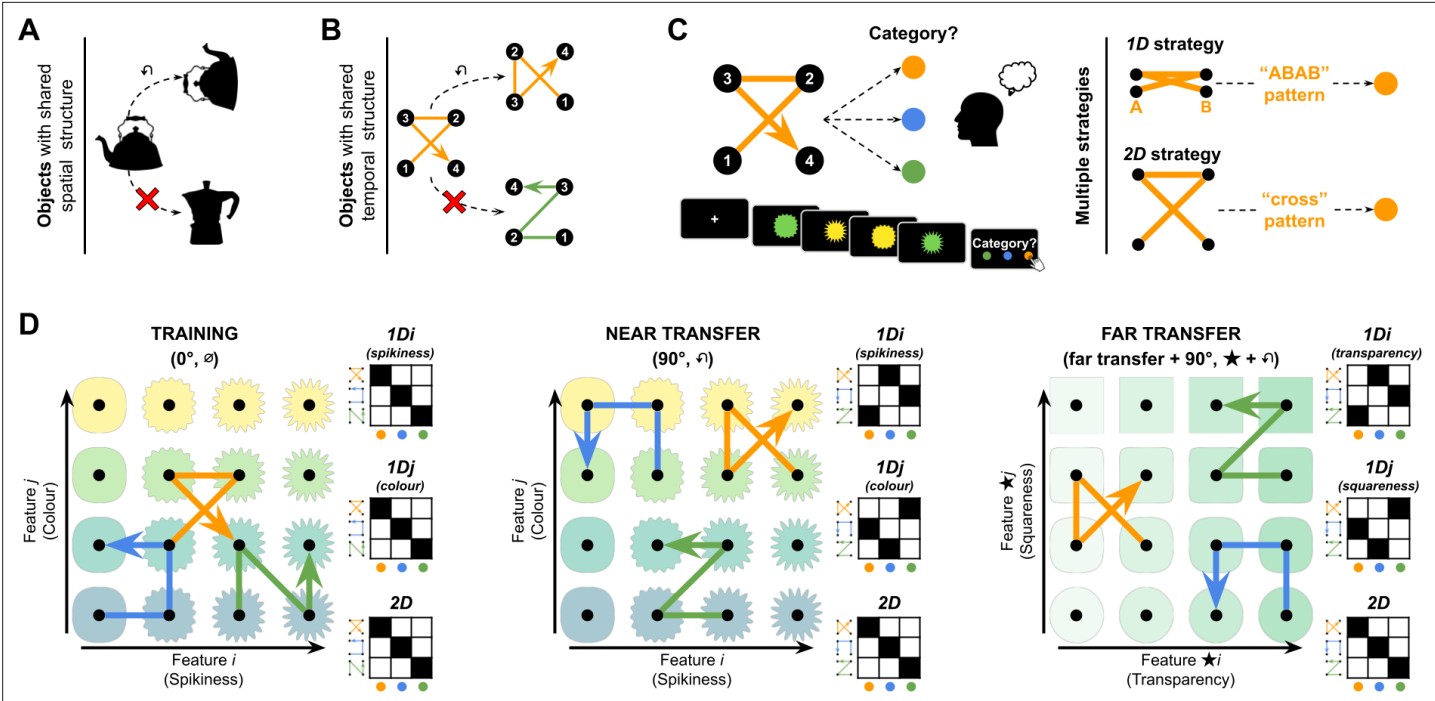

**Figure 1.** Paradigm. (**A**) Objects can be perceived as similar despite changes in shape, orientation, and features. (**B**) Similarly, can sequences of features be perceived as similar despite great changes in the features that composed them? (**C**) In our experiments, participants had to learn the category associated with quadruplets composed of four stimuli drawn from a 2D feature manifold. To do so, they could use one of two major strategies: tracking changes in a single dimension (*1D*) or two dimensions (*2D*). (**D**) Illustration of the feature manifold and transition vectors in the visual modality. Each transition vector defined transitions between cardinally or diagonally adjacent features on a 2D manifold (here, given by colour and spikiness for training and near transfer in the left and middle panels, and by transparency and squareness in the far transfer condition). The transition vectors for each category are shown in their canonical (0°) rotation as blue, orange, and green arrows superimposed on each feature manifold. The vectors are shown rotated by 90° in the near and far transfer conditions. Next to each feature manifold is a matrix showing the expected mapping (filled squares) from feature vectors to categories for a participant using the *1D* strategies (top and middle) and *2D* strategy (bottom). Note that the *2D* strategy always leads to effective generalisation (rotated exemplars are mapped on their corresponding categories) even in transfer, as indicated by filled squares on the matrix diagonal, whereas a *1D* strategy leads to a different pattern. We use the symbols ∅, ↻, and ★ to denote canonical (0°), 90° rotation, and far transfer conditions, respectively.

The online version of this article includes the following figure supplement(s) for figure 1:

**Figure supplement 1.** Examples of 2D feature manifolds used in the study.

**Figure supplement 2.** The eight transformations of the canonical quadruplets present in the study.

**Figure supplement 3.** Example trials.

**Figure supplement 4.** Model response matrices to the different quadruplets transformations in Experiment 1.

**Figure supplement 5.** Model response matrices to the different quadruplets transformations in Experiment 2, 3, and 4.

**Figure supplement 6.** Model recovery analysis.

## Results

On each trial, participants were presented with a sequence of four auditory, visual, or spatial stimuli (a quadruplet) drawn from 1 of 16 points on a continuously varying 2D (4×4) feature manifold. In the visual (auditory) modality, this manifold was respectively defined by two orthogonal and continuously varying visual (auditory) features. In the spatial domain, the 2D feature manifold was defined by positions in physical space, in either Cartesian or polar coordinates (see *Figure 1D*, *Figure 1—figure supplements 1–2*). Each quadruplet was constructed by first sampling a random point on the 2D feature manifold, and then iteratively choosing three further adjacent feature locations to make a sequence of four stimuli. In each experiment, there were three categories (*Figure 1C*). Each category was initially defined by a canonical set of transition vectors, which specified the three successive steps on the feature manifold (defining the positions from which stimuli in the quadruplet were sampled). Thus, for example, one set of transition vectors might be defined by compass directions {NE, W,

SE}. This would mean that after an initial stimulus was sampled, the second stimulus in the sequence would be the one NE in feature space, and the third W of that, and the fourth SE of that. We define *rotational generalisation* as the ability to recognise regularities in the sequence transition vectors that are independent of both translation and rotation. Thus, just as an upside-down teapot can still be recognised by the relative spatial relations among its handle, body, and spout, in Exp. 1 we asked whether concepts can be recognised when their associated transition vectors are rotated (e.g. vector sequence {NE, W, SE} on the feature manifold becomes {NW, S, NE} after 90° rotation). Note that in our study, quadruplets are also randomly translated on the manifold by virtue of the variable initial feature selection between trials. We thus make the basic assumption that rotational generalisation also involves translation invariance.

Our basic procedure was as follows. During training (120 trials), participants first learned to assign canonical (0° rotation) quadruplets to one of three categories using a button press, receiving fully informative feedback after each response (see *Figure 1—figure supplement 3*). Then, during test, participants performed a further 210 trials, half of which were identical to training (with feedback) while the other half were transfer trials involving categorisation of quadruplets whose feature transition vectors were rotated by 90°, 180°, or 270°. These novel quadruplets were either sampled from the same 2D feature manifold (e.g. colour and spikiness in the visual case; *near transfer* condition) or a new 2D feature manifold from the same modality (e.g. transparency and squareness; *far transfer* condition; see *Figure 1D*, *Figure 1—figure supplement 2*). Transfer trials received no feedback, allowing us to infer what knowledge was being generalised between training and transfer. Exp. 1–3 were pre-registered at https://osf.io/z9572/registrations.

## Concepts defined by spatial locations, but not auditory or visual features, are rotation-invariant

In Exp. 1, we recruited three cohorts of online participants (*N*=50 each, see *Figure 2*) to perform the task in the auditory, visual, and spatial modalities. These conditions differed only in how the feature manifold was defined: e.g., fundamental frequency and modulation frequency for auditory features (*Figure 2A*); e.g., spikiness and colour for visual features (*Figure 2E*); e.g., horizontal and vertical position for spatial locations (*Figure 2I*). Accuracy on training trials for each modality is shown in *Figure 2B, F, and J*. Participants learned the task well in all three conditions (but better in the spatial modality: intercept $\beta$=2.90 ± 0.19, slope associated with the auditory modality $\beta$=−1.91 ± 0.27, p<0.001, slope associated with the visual modality, $\beta$=−1.57 ± 0.26, p<0.001, mixed logistic regression on the probability of a correct response with participants as random effect). However, our main question was how participants would generalise learning to novel, rotated exemplars of the same concept.

To test this, we fit a family of quantitative models jointly to the training and transfer trials. To understand the logic of this modelling exercise, it is necessary to consider the alternative strategies that participants may have learned during training. Whereas rotation requires participants to represent both dimensions of the feature manifold (a rotation of 90° is only discernible in 2D), a viable alternative strategy during training is to base categorical decisions on a single feature (e.g. either spikiness or colour but not both). Each quadruplet consists of four adjacent feature locations forming a square on the feature manifold (*Figure 2D*) and thus the stimulation sequence comprises two features from each dimension. Thus, for example, if a participant attended only to spikiness, the four stimuli in a quadruplet would be represented as a feature pattern over spikiness levels (such as ABAB or ABBA, where A=more spiky, B=less spiky). During training, participants could learn to map these patterns onto categories, either in a signed fashion (e.g. ABAB maps to one category and BABA to another) or an unsigned fashion (ABAB and BABA both map to the same category). These strategies would lead to perfect performance during training, but would prevent the learning of rotational invariances. We built models that implemented these 1D strategies, which we call *1D_s* and *1D_u* respectively, and compared them to models that used both dimensions for categorisation (the *2D* model) or were simply responding randomly (*R* models). Each of these models predicts a unique pattern of generalisation (*Figure 1—figure supplements 4–5*) and only the *2D* model predicts that participants will assign rotated objects to the same category as their unrotated counterparts (rotational generalisation). Thus, the principal metrics we report in this study are the fraction of (non-random) subjects classified as *1D* vs. *2D* on transfer trials, which is a signature of whether the experimental conditions permitted the learning of rotational invariances for quadruplets. We report both fractions of participants (X/X best fit

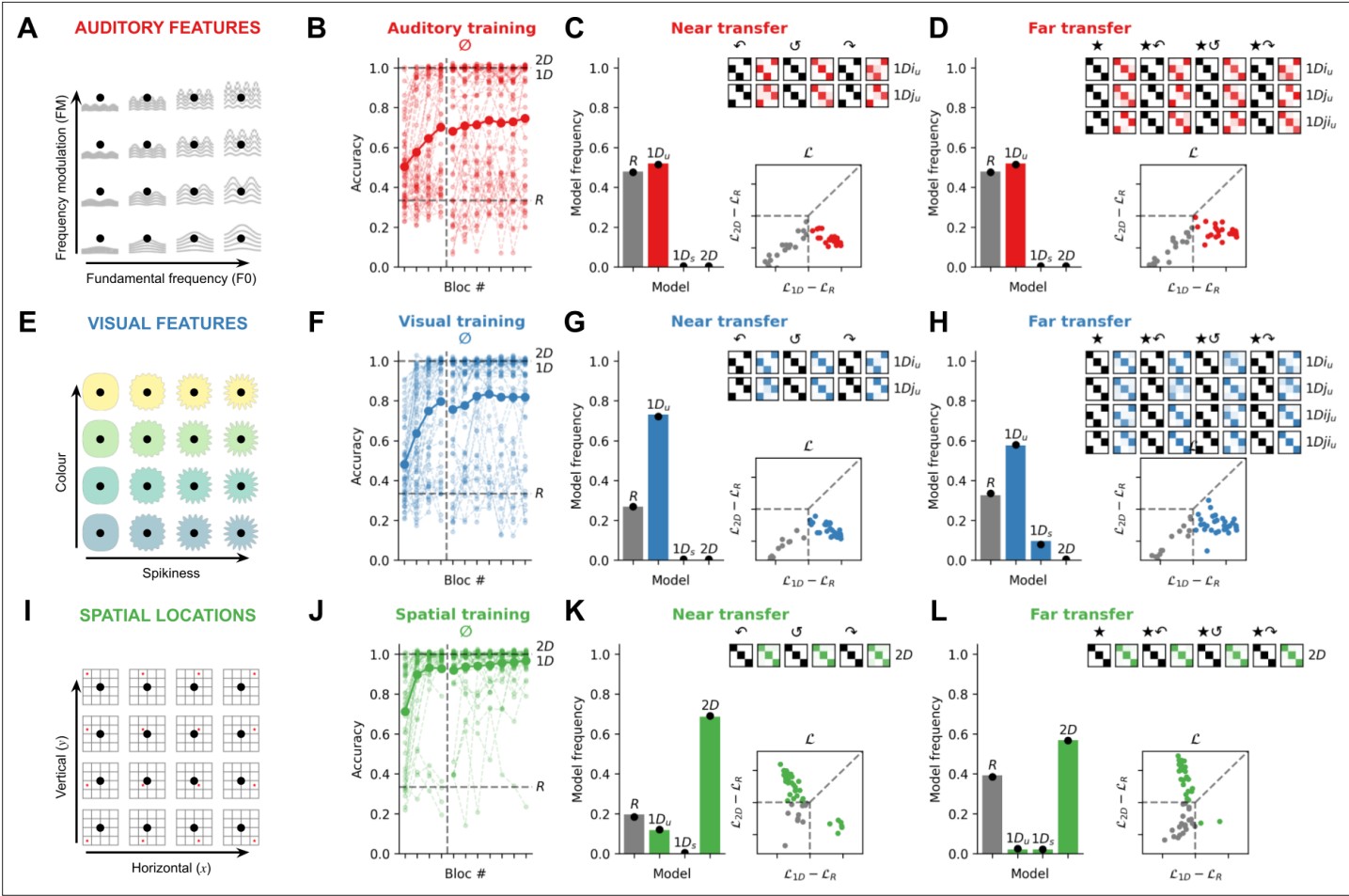

**Figure 2.** Experiment 1: Quadruplets composed of spatial locations, but not auditory or visual features, are associated with a *2D* strategy. (**A**) In the auditory modality (*N*=50), the 2D feature manifold was defined by auditory features. Here, we illustrate with a schematic in which the feature manifold is defined by fundamental frequency and frequency modulation. (**B**) Accuracy on training trials involving the canonical transition vectors (0° rotation, denoted by the symbol Ø). The bold curve and dots represent the group average; lighter curves are individual participants. During training, random models (*R*) are at chance (33% accuracy; lower dashed line), while idealised *1D* and *2D* models are at ceiling (100% accuracy; upper dashed lines). (**C**) Model fits to the near transfer responses. The bar plot shows model frequencies in the population and black dots are Bayesian point estimates of the model frequency. The matrices show cross-validated model predictions in the same format at *Figure 1D*, except with the degree of fading (light to dark colour) signalling average participant responses in each case. The matrices read as follows: the top row depicts the average behaviour of participants who were best fit by the *1Di_u* model (using a held-out set of responses). The *1Di_u* model predictions are shown in black for three rotations of the quadruplets (90°, 180°, 270° denoted by the symbols ↷, ↻, and ↶ respectively). The average response matrices of the participants using held-out responses are shown in red. The scatter plot below shows individual likelihoods for the *1D* and *2D* models (normalised by the *R* model). Each dot is an individual participant, coded by whether they are best fit by either *1D* or *2D* models (colour) or the *R* model (grey). Dashed lines distinguish zones of relative likelihood where participants are best fit by *R* (bottom left square), *1D* (rightmost quadrilateral), or *2D* (upper quadrilateral) models. (**D**) Model fits to the far transfer responses, using the same conventions as C. Note that, at transfer, more 1D models are possible because of the differing ways that participants could map between the *i* and *j* axes and the ★*i* and ★*j* axes during training and far transfer. (**E**) In the visual modality (*N*=52), the 2D feature manifold was defined by visual attributes; here we use colour and spikiness as an illustration. (**F–H**) depict data from the visual modality using conventions from **B–D**. (**I**) In the spatial modality (*N*=51), the 2D feature manifold was defined by the spatial location of a star (here a red dot is used for illustration). (**J–L**) depict data from the spatial modality using conventions from **B–D**.

by each model) and Bayes factors (BF) reflecting the relative likelihood of *1D* vs. *2D* models between conditions.

Consistent with our first pre-registered prediction, Exp. 1 revealed a striking dissociation in rotational generalisation between modalities. For near transfer, all non-random participants in the auditory and visual modality (26/26 and 38/38) learned a *1D_u* strategy, whereas the vast majority in the spatial modality (35/41) were best fit by a *2D* strategy. Bayesian group model comparison confirmed that the frequency of *1D* vs *2D* models among non-random participants was similar between the

auditory and visual modalities (BF=0.1, 'negative' evidence for a difference) but different between the auditory and spatial modalities (BF>100, 'decisive' evidence for a difference) and the visual and spatial modalities (BF>100; see Tables 6–9 for full results). This implies that the use of a 1D strategy (implying no rotational generalisation) was much more likely than a 2D strategy (implying rotational generalisation) when the manifold was defined by visual or auditory features (e.g. colour and shape or pitch and timbre), but the converse was true when the feature manifold was defined by coordinates in physical space (e.g. horizontal and vertical position).

For far transfer, the results were very similar. In the auditory modality, all non-random participants (26/26) were again best fit by a *1D_u* strategy, and in the visual modality, most (30/35) were fit by a *1D_u* strategy, 5/35 by a *1D_s* strategy, and none by a *2D* strategy (difference between auditory and visual, BF = 0.1). By contrast, in the spatial modality, where far transfer involved remapping from cardinal to polar coordinates or vice versa, almost all non-random participants (29/31) were again best fit by a *2D* strategy (both BF>100 comparing with the auditory and visual modalities, 'decisive' evidence for a difference). Behaviour in each modality of Exp. 1 is illustrated in *Figure 2*, where we display category assignments under each rotation for participants allocated to distinct model classes on the basis of

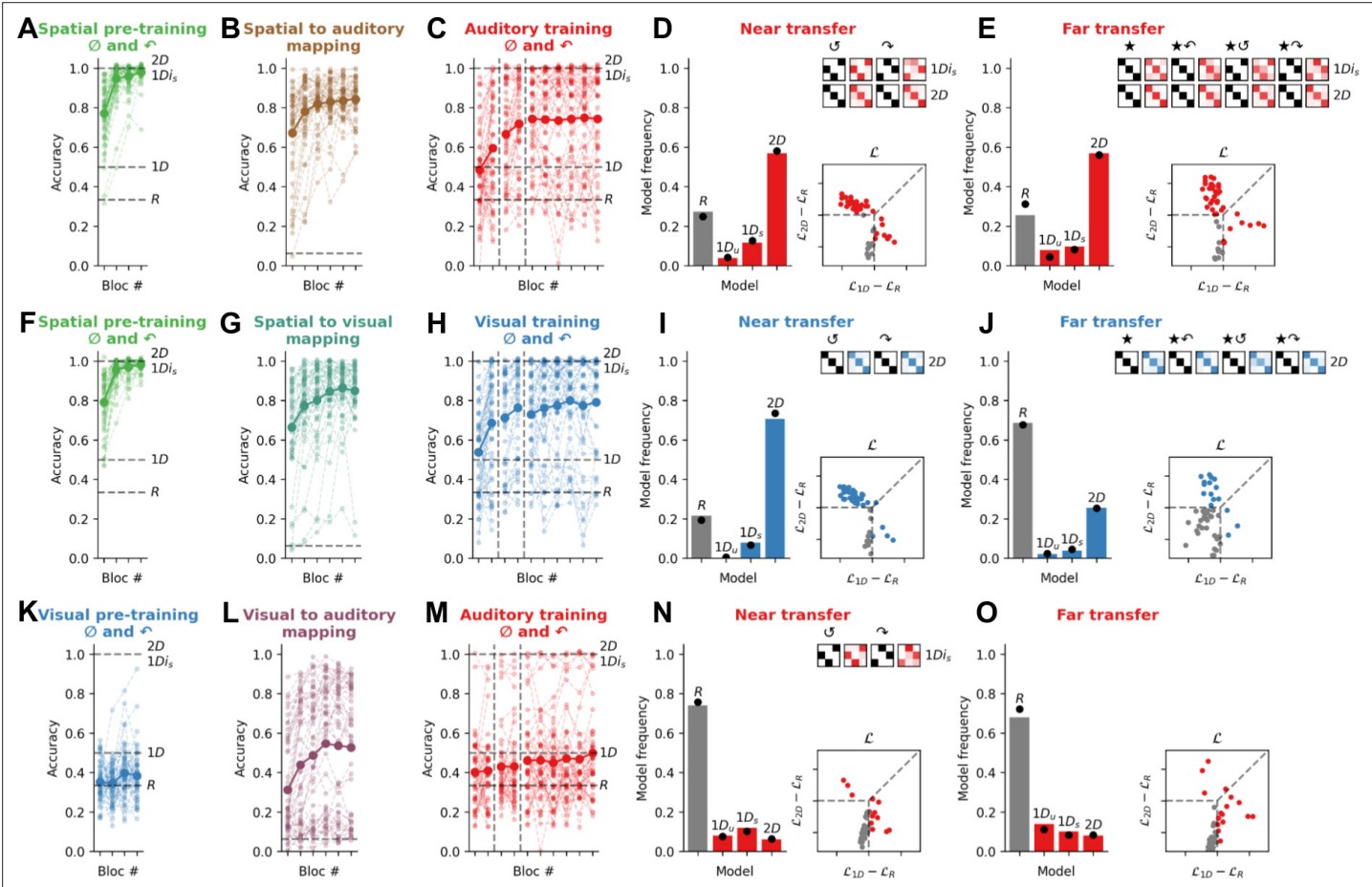

**Figure 3.** Experiment 2: Spatial pre-training triggers the use of a *2D* strategy for quadruplets composed of auditory or visual features. (**A**) Performance on the spatial pre-training task in Exp. 2a (*N*=51). For this and all plots below, the bold line is the group average and lighter lines are individual participants. Dashed lines show expected average performance under the corresponding models (labelled). (**B**) Performance in the spatial to auditory mapping task. Chance performance is at 1/16 (dashed line). (**C**) Performance on training trials in the auditory modality for canonical (0°) and 90° rotated quadruplets. During training, random models (*R*) are at chance (33% accuracy), almost all *1D* models are at 50% accuracy, and *1D_i_s* and *2D* models are at ceiling (100% accuracy). (**D**) Model fits to the near transfer responses, using the same conventions as *Figure 2*. Model predictions are shown in black for two rotations of the quadruplets (180° and 270° denoted by the symbols ↺ and ↻, respectively). (**E**) Model fits to the far transfer responses. (**F–J**) read as A–E for Exp. 2b (*N*=51), where participants performed a spatial pre-training and a spatial to visual mapping task prior to performing the visual version of the main task. (**K–O**) read as A–E for Exp. 2c (*N*=50), where participants performed a visual pre-training and a visual to auditory mapping task prior to performing the auditory version of the main task.

held-out data. Together, these data show definitively that, when categories were characterised by temporal patterns in spatial location (e.g. where transitions in physical space were aligned with those on the feature manifold), participants learned to represent the *2D* structure of the concept, and generalised readily to rotated (as well as translated) exemplars. However, when concepts were defined by patterns of nonspatial auditory or visual features, participants learned mappings to each category by relying on a single feature dimension and thus failed to form rotational invariant representations.

## Spatial pre-training provides a scaffold for rotational generalisation in the auditory and visual modalities

Exp. 1 shows that rotational generalisation succeeds for spatial concepts but fails for nonspatial concepts. Next, in Exp. 2 we tested our main prediction: that space can be used as a scaffold for rotational generalisation of nonspatial concepts. We recruited three new cohorts of participants (*N*=50 each, see *Figure 3*) to perform a multi-phase task that unfolded over 2 successive days. On day 1, participants received 60 pre-training trials in the *pre-training* modality. These trials matched training trials in Exp. 1 for the corresponding modality (spatial or visual) except that they comprised both canonical (0°) and 90° rotated quadruplets, but not those rotated by 180° or 270° (we included examples of rotated quadruplets in the training set to encourage rotational generalisation, but as shown in Exp. 3, results do not depend on this choice). Subsequently, participants performed 288 trials of a multimodal association task, in which they learned the association between each of the 16 stimuli in the pre-training modality and their corresponding stimulus in a different *testing* modality, where the corresponding stimulus occupied an equivalent position on the 2D feature manifold (we call this the 'mapping task'). The goal of this task was to teach participants' correspondences between either spatial and visual, spatial and auditory, or visual and auditory feature manifolds. Then on day 2, after some refresher pre-training and mapping trials, participants performed the same task as in Exp. 1 in the testing modality, again with the exception that training trials also included 90° rotated quadruplets.

Our pre-registered prediction for Exp. 2 was that when (Cartesian) physical space was the pre-training modality, participants would now (in contrast to Exp. 1) learn using a predominantly *2D* strategy in both auditory (Exp. 2a) and visual (Exp. 2b) testing modalities. In other words, by learning the association between auditory or visual features and a corresponding spatial location, concepts composed of exclusively nonspatial features could now be generalised over rotations in a way not exhibited by a single participant in Exp. 1a or Exp. 1b. By way of control, however, we predicted that when the pre-training involved (nonspatial) visual features, no such benefit would occur, and participants would fail to show rotational invariance.

This is exactly what we found, for both near and far transfer. In the near transfer condition, with spatial pre-training, 29/37 non-random participants were best fit by a *2D* strategy when audition was the testing modality, and 36/40 when vision was the testing modality. By contrast, however, participants who underwent visual (rather than spatial) pre-training failed to show a benefit when audition was the testing modality. In fact, most (37/50) were best fit by the random model (see below), with 6/13 non-random participants favouring a $1D_s$ strategy. We once again calculated BF at the group level to assess the reliability of these results. We found that BFs exceeded 100, providing 'decisive' evidence that the *2D* model was more favoured among the groups with spatial pre-training than that without. Similarly, in the far transfer condition, spatial pre-training allowed 29/38 participants in the auditory modality and 13/16 participants in the visual modality to successfully generalise via a *2D* strategy. This was not the case for participants who experienced visual pre-training (again, frequency of *1D* vs *2D* models between conditions: BF>100, 'decisive'). In other words, spatial pre-training provided an effective scaffold that allowed participants to learn auditory and visual objects in a 2D representational format that permitted generalisation to novel rotated exemplars.

Participants in Exp. 2c performed poorly during training and were more likely to be fit by the random models during transfer than those who performed the same auditory task in Exp. 1a. Indeed, we computed the BF quantifying the relative likelihood of the random (*R* models) vs all other models (*1D* and *2D* models) and found 'substantial' evidence in favour of a difference between groups both in near transfer (BF = 7.4) and in far transfer (BF = 2.7). This might seem curious, because Exp. 2 participants had access to more diverse training (on both 0° and 90° quadruplets) as well as the supplementary visual pre-training. Why did Exp. 2c participants struggle with the task? In fact, this phenomenon

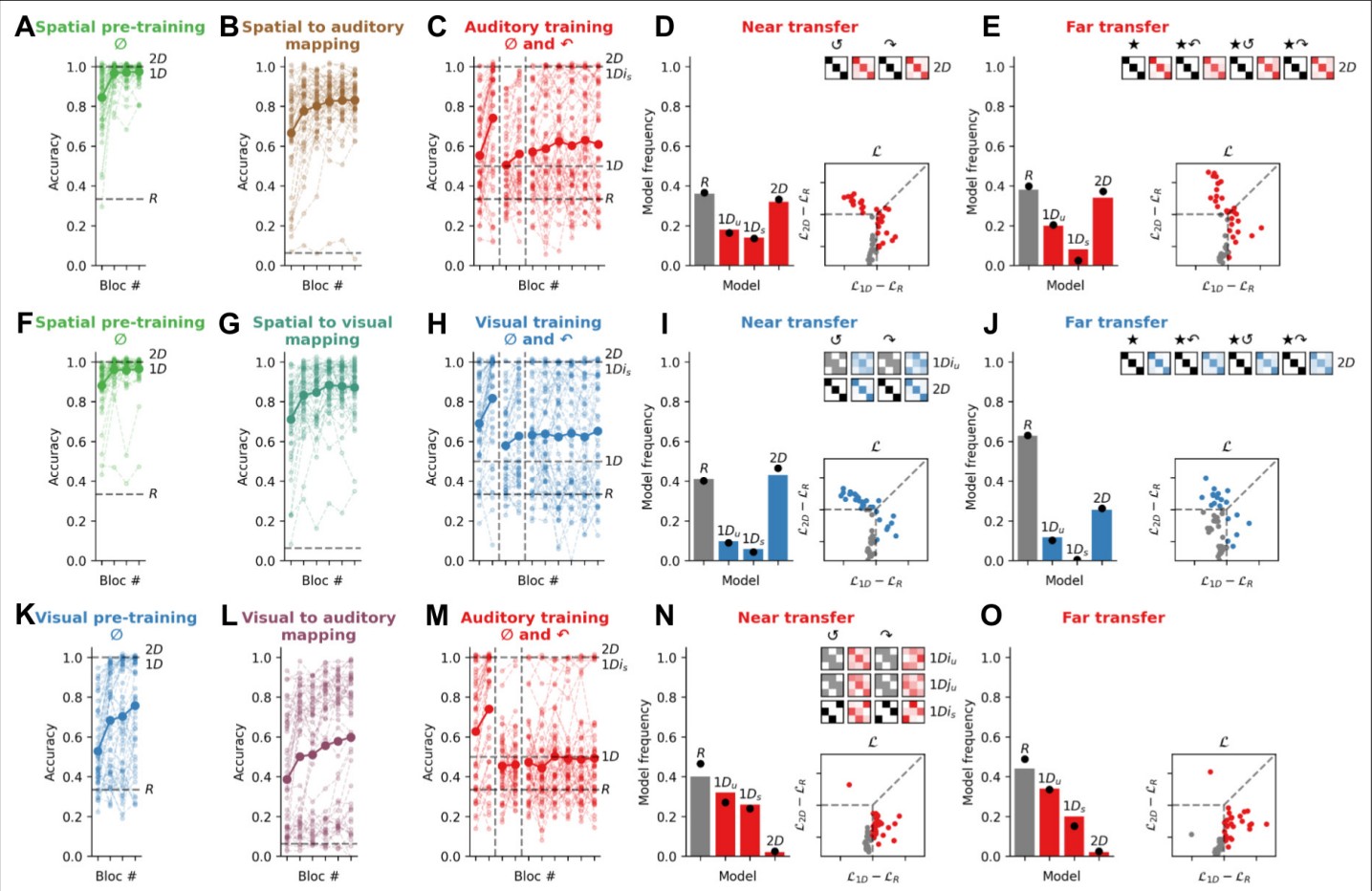

**Figure 4.** Experiment 3: Exposure to the canonical events (0°) during spatial pre-training is sufficient to trigger the use of a *2D* strategy. All panels use the same conventions as *Figure 3*. (**A–E**) Contrary to Exp. 2a, in Exp. 3a (*N*=50), participants were exposed only to canonical events (0°) during the spatial pre-training. (**F–J**) Exp. 3b (*N*=51). (**K–O**) Exp. 3c (*N*=50).

The online version of this article includes the following figure supplement(s) for figure 4:

**Figure supplement 1.** Experiment 4: (**A–E**).

makes sense, because training with feedback on both 0° and 90° quadruplets effectively invalidates a *1D* strategy, because there no longer exists a unique mapping between categories and features in either of the two feature dimensions (note that training performance in Exp. 2c plateaus close to 50%). This lack of a viable *1D* strategy during training obliges participants to use a *2D* strategy where possible. Because this is only possible with spatial pre-training, in Exp. 2c they revert to random. Whilst this explains what we observed in Exp. 2, it also allows a further prediction: if we remove the 90° rotated quadruplets from pre-training, then participants in the spatial pre-training modality should be somewhat less prone to use a *2D* strategy (because *1D* is available) whereas participants who undergo visual pre-training should show more *1D* behaviour at the expense of the random model. In Exp. 3, we tested and confirmed this prediction.

Exp. 3 involved three new cohorts (*N*=50 each, see *Figure 4*) and was identical to Exp. 2, except that now pre-training trials consisted exclusively of canonical (0°) quadruplets (although 90° quadruplets were still present when the testing modality was trained on day 2). As predicted, non-random participants who enjoyed spatial pre-training were still prone to use a *2D* strategy when audition was the testing modality (16/32 for near transfer and 17/31 for far transfer) as well as when vision was the testing modality (22/30 and 13/19), replicating the findings of Exp. 2. However, compared to Exp. 2, overall more participants relied on *1D* strategies. In the auditory modality in Exp. 3a, 16/32 were best fit by a *1D* model in the near transfer condition (9/32 $1D_u$ and 7/32 $1D_s$) and 14/31 in the far transfer condition (10/31 $1D_u$ and 4/31 $1D_s$). At the group level, the BF confirmed that participants were more

likely to be fit by a *1D* model in Exp. 3a than Exp. 2a in the auditory modality (frequency of *1D* vs *2D* models between Exp. 2a and Exp. 3a, near transfer BF = 3.4 'substantial' evidence, far transfer BF = 1.5 'weak' evidence). Similarly, in the visual modality in Exp. 3b, 8/30 were best fit by a *1D* model in the near transfer condition (5/30 *1D$_u$* and 3/30 *1D$_s$*) and 6/19 in the far transfer condition (6/19 *1D$_u$*) (again, frequency of *1D* vs *2D* models between Exp. 2b and Exp. 3b, near transfer BF = 5.9 'substantial' evidence, far transfer BF = 2.2 'weak' evidence). By contrast, participants who underwent nonspatial (visual) pre-training did not use a *2D* strategy (1/30) but rather preferred *1D* strategies in both near transfer (13/30 *1D$_u$* and 16/30 *1D$_s$*) and far transfer conditions (10/28 *1D$_u$* and 17/28 *1D$_s$*). Comparing these results with the frequency of *1D* vs *2D* models in conditions with spatial pre-training (Exp. 3a and 3b), we found that all BFs exceeded 100, providing 'decisive' evidence that the *2D* model was more favoured among the groups with spatial pre-training than that without.

Thus, these results show that training exclusively on canonical (0°) quadruplets facilitates a 1D strategy, which is expressed more readily than in Exp. 2; but that the 2D strategy is still more likely for participants who underwent spatial pre-training. Further, the results show that participants who did not experience spatial pre-training were still engaged in the task, but were not using the same strategy as the participants who experienced spatial pre-training (1D rather than 2D). Thus, the benefit of the spatial pre-training is not simply to increase the cognitive engagement of the participants. Rather, spatial pre-training provides a scaffold to learn rotation-invariant representation of auditory and visual concepts even when rotation is never explicitly shown during pre-training. Furthermore, participants

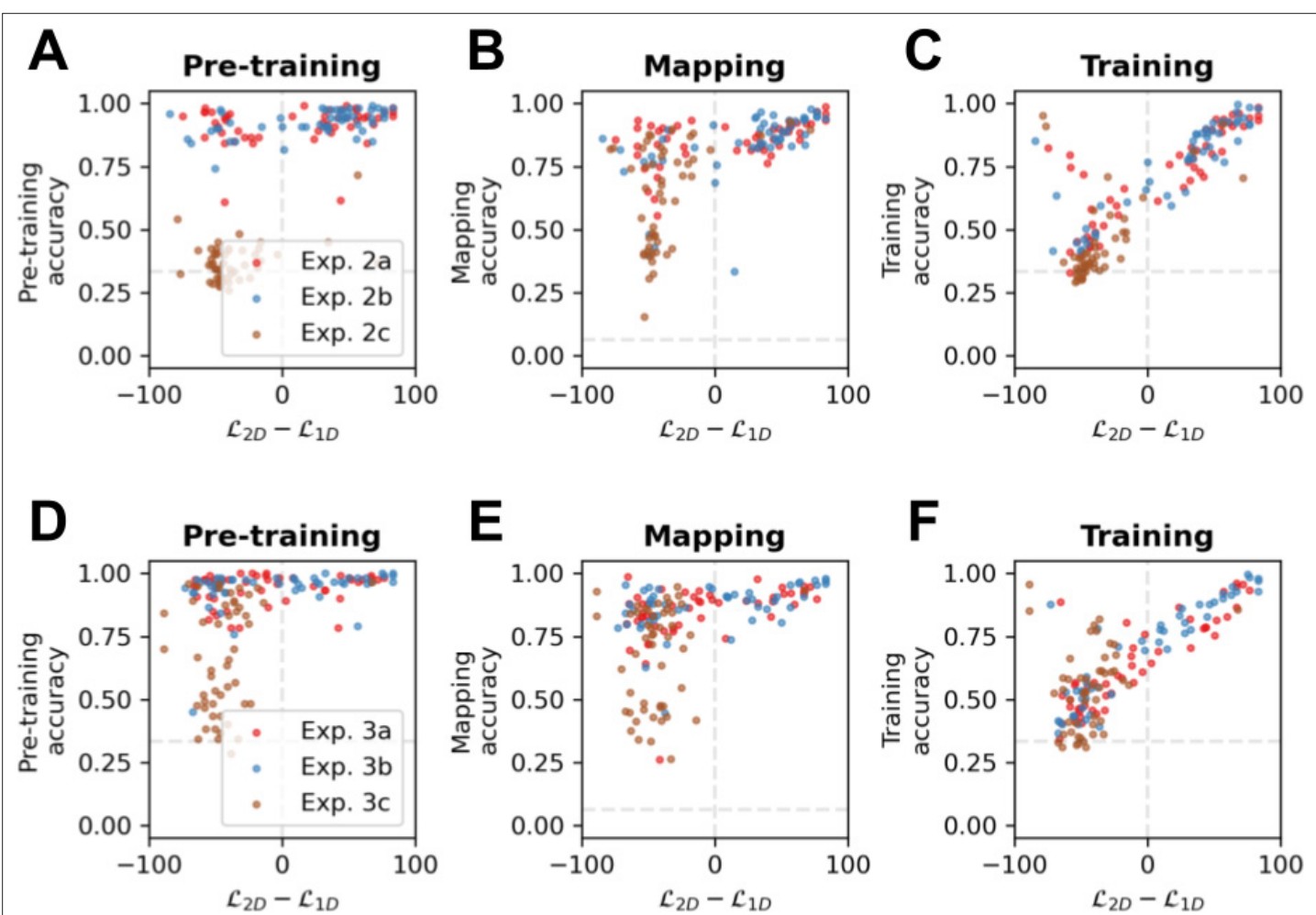

**Figure 5.** Correlation analysis. Scatter plots of *2Dness* (the difference in likelihood between the best *1D* model and the *2D* model in the training modality in near transfer) and (**A**) pre-training accuracy, (**B**) mapping accuracy, and (**C**) training accuracy in the testing modality in Exp. 2. Dots are individual participants in Exp. 2a (red), 2b (blue), and 2c (brown). (**D–F**) read as **A–C** for Exp. 3.

are sensitive to the available strategies during pre-training, and use the *1D* strategy when possible if they have not learned to associate features with space.

## Spatial mapping performance predicts rotational generalisation for nonspatial modalities

Next, we used our data from Exp. 2 and Exp. 3 to study how performance on each phase of our task predicted rotational generalisation in the testing phase (see *Figure 5*). For each participant, we created an index of rotational generalisation (*2Dness*) as the difference in log-likelihood between the best *1D* model and the *2D* model during near transfer. We found that *2Dness* was powerfully predicted by training accuracy (Pearson correlation between *2Dness* and training accuracy [$r_{2Dness,TRAINING}$] = 0.80, p<0.001) in both Exp. 2 ($r_{2Dness,TRAINING}$ = 0.83, p<0.001) and Exp. 3 ($r_{2Dness,TRAINING}$ = 0.78, p<0.001). The fact that training performance is highly correlated with *2Dness* implies that participants who solved the training task formed representations that were generalisable in 2D; in other words, very few participants overfit to the training set. Accordingly, participants were poorly captured by an additional model (the *R'* model; 4/152 in Exp. 2, 4/151 in Exp. 3), that has perfect performance during training but responds randomly during transfer. Next, we asked whether accuracy during pre-training and mapping were systematically associated with *2Dness*, and assessed their relative importance using partial correlations. Pre-training did explain unique variance in *2Dness* after accounting for mapping (correlation between pre-training and *2Dness* after partialling out mapping [$r_{2Dness,PRETRAINING - MAPPING}$]=0.17, p<0.01) and vice versa ($r_{2Dness,MAPPING - PRETRAINING}$ = 0.27, p<0.001). However, *2Dness* was better predicted by mapping than by pre-training in Exp. 2a ($r_{2Dness,MAPPING - PRETRAINING}$ = 0.42, p<0.005 and $r_{2Dness,PRETRAINING - MAPPING}$ = 0.30, p<0.05), Exp. 2b ($r_{2Dness,MAPPING - PRETRAINING}$ = 0.41, p<0.005 and $r_{2Dness,PRETRAINING - MAPPING}$ = 0.04, p=0.81), Exp. 3a ($r_{2Dness,MAPPING - PRETRAINING}$ = 0.32, p<0.05 and $r_{2Dness,PRETRAINING - MAPPING}$ = 0.06, p=0.70) and Exp. 3b ($r_{2Dness,MAPPING - PRETRAINING}$ = 0.46, p<0.001 and $r_{2Dness,PRETRAINING - MAPPING}$ = 0.33, p<0.05). The strong correlations between the mapping task performances and *2Dness* suggest that learning the association between nonspatial and spatial features is the critical step that allows rotational generalisation.

We tested and confirmed this prediction in Exp. 4 (see *Figure 4—figure supplement 1*) which repeated Exp. 3 except that spatial pre-training was replaced with a duration-matched filler task (in which the category is defined by the number of stationary blue stars in a sequence). Without spatial pre-training, a sizeable proportion of participants still learned a *2D* strategy in both the auditory (9/30 in near transfer, 9/28 in far transfer) and visual (12/19 and 7/19) modality, although the majority relied on a 1D strategy (auditory modality: 4/30 *1D$_u$* and 17/30 *1D$_s$* for near transfer, 4/38 *1D$_u$* and 15/28 *1D$_s$* for far transfer; visual modality: 5/19 *1D$_u$* and 6/19 *1D$_s$* for near transfer, 5/14 *1D$_u$* and 2/14 *1D$_s$* for far transfer). In the auditory modality (Exp. 4a), this can be compared with Exp. 2c, where almost all participants were using a random strategy (frequency of *R* vs *1D/2D* models, BF = 34.0, 'strong' evidence), and with Exp. 3c where almost no participants were using a *2D* strategy (frequency of *1D* vs *2D* models, BF = 8.6, 'substantial' evidence). Thus, for ~20% participants, the mere exposure to the mapping was sufficient to benefit from the spatial scaffolding effect and actually seeing the quadruplets in the spatial modality was not necessary for them.

## Discussion

We studied the conditions under which participants learn rotation- and translation-invariant representations of abstract concepts. We found that participants can generalise conceptual knowledge to novel sequences (quadruplets) defined by rotations of stimulus feature transition vectors, but only if the features were themselves physical spatial locations (e.g. *x, y* position; Exp. 1) or if nonspatial attributes had previously been mapped to a physical spatial location in a pre-training task (Exp. 2–4). Thus, an explicit representation of physical space is a 'scaffold' that permits objects to be learned in a rotational-invariant fashion, and thus allows rotational generalisation. This supports the idea that neural representations of space form a critical substrate for learning abstractions in nonspatial domains (*Behrens et al., 2018*; *Bellmund et al., 2018*; *Summerfield et al., 2020*; *Gärdenfors, 2014*).

It is well known that humans learn rotational invariances for visual objects, whose features are organised in physical space. For example, an upside-down teapot can be recognised by the relative position of handle, lid, and spout. This case mimics our spatial modality condition, where each

concept was a pattern of locations in physical space. It is thus perhaps unsurprising that rotational generalisation is possible in this condition. However, we found it striking that participants generalised in such different ways when the features in question were drawn from a nonspatial manifold, in either the visual or auditory domain. In these conditions, participants seemed to have no trouble recognising patterns that were consistently translated in feature space. This is consistent with previous studies that have shown that we can understand language in different accents, or name a familiar tune played at an atypical speed or pitch (*Dupoux and Green, 1997*). However, they did so via a representation that focused on just one of the two possible dimensions, and thus did not permit rotational generalisation. There was thus a clear dissociation between human ability to generalise patterns in physical space and a more abstract feature space.

Next, we showed that spatial pre-training allowed rotational generalisation even for sequences composed of nonspatial features. This implies that the neural representation of space may serve as a 'scaffold', allowing people to visualise and manipulate nonspatial concepts. One alternative explanation of this effect could be that the spatial pre-training encourages participants to attend to both dimensions of the nonspatial stimuli. By contrast, pre-training in the visual or auditory domains (where multiple dimensions of a stimulus may be relevant less often naturally) encourages them to attend to a single dimension. However, data from our control experiments, Exp. 2c and Exp. 3c, are incompatible with this explanation. Around ~65% of the participants show a level of performance in the multimodal association task (>50%) which could only be achieved if they were attending to both dimensions (performance attending to a single dimension would yield 25% and chance performance is at 6.25%). This suggests that participants are attending to both dimensions even in the visual and auditory mapping case. Rather, whilst we are not aware of previous studies that have tested spatial scaffolding in the way described here, our findings are consistent with the more general idea that space is represented in an overlapping fashion with nonspatial information, such as time or number (*Dehaene et al., 2022*). For example, sequences with regular spatial geometry are learned more readily than those composed of arbitrary patterns (*Al Roumi et al., 2021*). Our findings also cohere with evidence that visuospatial skills are correlated with a variety of academic competences, especially in STEM subjects such as maths and engineering (*Stieff and Uttal, 2015*), and that spatial training interventions (such as teaching mental rotation) in educational settings can improve nonspatial abilities, such as calculus grades (*Sorby et al., 2013*).

The idea that spatial representations form a generalised substrate for cognition – including for coding temporal structure – draws on a long tradition in philosophy (*Kant, 2007*), cognitive science (*Gärdenfors, 2000*), and neuroscience (*Behrens et al., 2018*; *Bellmund et al., 2018*; *Summerfield et al., 2020*). The precise substrate for this effect is unclear, but it seems likely that neural assemblies activated by physical locations in space (e.g. in parietal or medial temporal lobe areas) are recycled for representing nonspatial patterns in data. We acknowledge that our study does not provide a mechanistic model of the spatial scaffolding effect but rather delineate which aspects of the training are necessary for generalisation to happen. In our study, thus, the mapping task facilitates this recycling by teaching participants a point-to-point mapping between nonspatial feature combinations and locations in physical space. Indeed, our correlation analysis and Exp. 4 suggested that successfully learning mappings between spatial and nonspatial features was the strongest determinant of rotational generalisation. This mapping task was presented in an egocentric frame of reference defined by the $x$, $y$ coordinates of the screen. Explicit representations of location in egocentric space in the primate are found in dorsal stream structures such as the posterior parietal cortex (*Colby et al., 1993*). Current deep networks – which successfully categorise lone objects in a natural image but often fail on tests of relational reasoning or scene understanding – may be hampered by their failure to represent space explicitly in this way (*Summerfield et al., 2020*).

All the effects observed in our experiments were consistent across near transfer conditions (rotation of patterns within the same feature space), and far transfer conditions (rotation of patterns within a different feature space, where features are drawn from the same modality). This shows the generality of spatial training for conceptual generalisation. This means that an explicit representation of space might be the substrate for strong forms of transfer observed in humans, such as when we understand the shared meaning between 'red, amber, green' at a traffic light and 'ready, set, go' before a race. We did not test transfer across modalities nor transfer in a more natural setting; we leave this for future studies.

**Table 1.** Demographic data.

$\mu$: average, $\sigma$: standard deviation, F: female, M: male.

|  | **N** | **Age ($\mu \pm \sigma$)** | **Sex (F/M/other)** |
|---|---|---|---|
| Exp. 1 | 153 | 29.4±6.1 | 70/68/15 |
| Exp. 2 | 152 | 29.9±5.9 | 88/62/2 |
| Exp. 3 | 151 | 29.4±5.4 | 88/63/0 |
| Exp. 4 | 102 | 28.5±6.1 | 65/37/0 |
| Total | 558 | 29.4±5.9 | 311/230/17 |

# Materials and methods

Exp. 1, 2, and 3 and analyses were pre-registered. The pre-registration documents can be found at https://osf.io/z9572/registrations.

## Stimuli and paradigm

### Participants

In total, we collected data from 558 participants with the following demographic characteristics (see *Table 1*).

Participants were recruited on the crowdsourcing platform Prolific (https://app.prolific.co/). Inclusion criteria included being between 18 and 40 years of age, reporting no neurological condition, being an English speaker, being located in the USA or the UK, not having participated in another version of the task, having a minimal approval rate of 90% on Prolific, and having a minimum of five previous submissions on Prolific. Participants received on average £10/hr for their time and effort, including a bonus on performance (£8.5/hr with random performances, £10.5/hr with perfect performances). All experiments were approved by the Medical Sciences Research Ethics Committee of the University of Oxford (approval reference R50750/RE005). Before starting the experiment, informed consent was taken through an online form, and participants indicated that they understood the goals of the study, how to raise any questions, how their data would be handled, and that they were free to withdraw from the experiment at any time.

The sample size was determined prior to the data collection, as indicated in the pre-registration documents.

### Stimuli

Across all experiments, we presented sequences of four stimuli ('quadruplets'). The stimuli occurred in one of three modalities: auditory, visual, or spatial. The quadruplet consisted of four successive auditory, visual, or spatial features, each drawn from 1 of 16 points (arranged in a 4×4 grid) on a 2D feature manifold ($i$, $j$). The dimensions of the manifold differed as a function of the modality, with four stimulus dimensions per modality (see *Figure 1—figure supplement 1*). For each participant, given the relevant modality, two stimulus dimensions were randomly selected to form the dimensions of the original manifold (for training and near transfer; denoted by $i$, $j$) and the two other dimensions were selected to form the dimensions of the far transfer manifold (for far transfer; denoted by ★$i$, ★$j$). In each experiment, the stimulus dimensions assigned to the $i$ and $j$ dimension of the original manifold and the ★$i$ and ★$j$ dimensions of the far transfer manifold were randomised across participants.

In the auditory modality, stimuli were 500 ms complex modulated tones generated with the *sndlib* module of the *pychoacoustics* Python library (version 0.4.6, https://pychoacoustics.readthedocs.io/), with the following features:

- Fundamental frequency F0 (110, 220, 330, or 440 Hz)
- Frequency modulation FM (1, 2, 3, or 4 Hz)
- Amplitude modulation AM (1, 2, 3, or 4 Hz)
- Number of high harmonics (1, 3, 7, or 10).

Any combination of two features could be chosen as manifold feature dimensions except the combination FM and AM, because it is perceptively hard to discriminate FM and AM in a single sound.

In the visual modality, stimuli were Fernandez-Guasti squircle presented on a black background, generated with the *matplotlib* Python library (version 3.6.2, https://matplotlib.org/), with the following features:

- Colour (viridis perceptually uniform colormap, 0, 0.33, 0.66, or 1)
- Transparency level (alpha level, 0.2, 0.46, 0.73, or 1),
- Squareness (squareness parameter of the Fernandez-Guasti squircle, 0.01, 0.8, 0.98, or 1)
- Spikiness (amplitude of the cosine modulation relative to the squircle radius, 0, 0.06, 0.13, or 0.2).

Any combination of two features could be chosen as manifold feature dimensions except the combination transparency level and colour, because it is perceptively hard to discriminate the level of transparency and colour in a single image.

In the spatial modality, stimuli were a red star with different spatial locations presented on a black background, also generated with *matplotlib*, with the following features:

- Horizontal position (1, 2, 3, or 4)
- Vertical position (1, 2, 3, or 4)
- Radius (1, 2, 3, or 4)
- Polar angle (0, 90°, 180°, or –90°).

Horizontal position and vertical position, as well as radius and angle, were systematically associated. This is because the other feature combinations, such as radius and horizontal position, are impossible.

The precise intensity level of the auditory stimuli and the precise size of the visual stimuli were dependent on the participant's headphones and screen and are thus unknown.

## Procedure

### JavaScript online experiment

The experiment was written in JavaScript, using *jsPsych* (version 7.3.1, https://www.jspsych.org/7.3/) (*de Leeuw, 2015*), and hosted on a web server. Scripts are available at https://osf.io/z9572.

### Game design

The whole experiment was presented to the participants as an 'interstellar mission' game. The goal of this 'interstellar mission' was to establish contact with aliens on a distant planet. In the main task, participants were asked to 'identify the aliens on the planet by paying attention to the sequence that they produce'. In the mapping task, participants were asked to 'associate each alien sound (/image) with a spatial location (/image) on the screen'.

### Screening task

A screening task was performed prior to the experiment to ensure that the auditory conditions under which recruited participants performed the experiment were sufficient to discriminate the sounds, and to verify that participants were able to pay attention to a complex cognitive task. The screening task was an 8 min long, 2-back auditory task. Stimuli were artificially generated impact sounds of wood, metal, and glass (*Aramaki et al., 2006*). All sounds had the same fundamental frequency, loudness, and duration, and differed only in timbre (examples of 'tuned' sounds available at http://www.lma. cnrs-mrs.fr/~kronland/Categorization/sounds.html). Each sound was 400 ms long, with cosine ramp on and off of 10 ms. Trials consisted of the following events: (1) sound presentation for 400 ms, (2) key press recording for 1000 ms, (3) trialwise feedback for 800 ms, and (4) an inter-trial interval for 1000 ms (in total, 3200 ms per trial). On every trial but the first two, participants had to indicate whether the sound was the same as the sound presented two trials before, by pressing a key on the keyboard (key [S] for 'same' and key [D] for 'different'). Participants received feedback on every trial. 150 trials were presented. Participants reaching 75% accuracy were recruited in the main experiment. This corresponded to ~40% of participants. Batches of 100–250 participants were screened and allocated to one experiment and one condition until the desired sample size was reached for all experiments. All participants in all experiments did the screening task prior to the experiment.

### Main task

In the main task, participants were asked to infer the category of a quadruplet consisting of four successive visual, auditory, or spatial features (see *Figure 1—figure supplement 3A*). There were

three possible categories. Each category was defined by a canonical set of transition vectors, which specified three successive steps in a 2D feature manifold (category 0: {E, N, W}, category 1: {NE, W, SE}, category 2: {N, SE, N}). The quadruplets were further rotated and embedded in either the original manifold or the far transfer manifold, leading to eight transformations: canonical (∅), 90° rotation (↷), 180° rotation (↻), 270° rotation (↶), far transfer canonical (★), far transfer 90° rotation (★ + ↷), far transfer 180° rotation (★ + ↻), and far transfer 270° rotation (★ + ↶) (see *Figure 1—figure supplement 2*). Trials consisted of the following events: (1) a black loading screen for 500 ms, (2) quadruplet presentation for 8000 ms (four times 500 ms of stimulus presentation followed by 1500 ms black screen), (3) response recording window until a response was made, and (4) trialwise feedback for 800 ms. For trials without trialwise feedback, a black screen was presented for 800 ms instead of the feedback screen. Response was made by clicking with the mouse on one of three buttons that appeared on screen. The ordering of the buttons was randomised across participants, and kept fixed for the entire experiment. The ordering of the trials was pseudo-randomised such that exemplars from each of the three categories appeared 10 times each every block (30 trials). The starting location for the transition vector on the feature manifold was chosen randomly every trial from among nine possible positions (excluding the outer ring). Participants were instructed that the task was deterministic. ('*The rules used by the aliens to produce the sequences are 100% deterministic. This means that once you have discovered the rules, you will reach 100% of correct responses*'.) On top of trialwise feedback on training trials, participants received blockwise feedback on their performance in the last block. Trials without trialwise feedback were not used to compute this blockwise feedback. See below for the exact trial numbers and ordering.

## Mapping task

In the mapping task, participants had to learn associations between features from different modalities (see *Figure 1—figure supplement 3B*). When space (/vision) was the pre-training modality and auditory (/visual) the testing modality, on each trial participants learned to associate one auditory (/visual) feature with its corresponding (spatial/visual) feature. For the spatial domain, this means mapping position on the latent manifold ($i$, $j$) onto its corresponding location in physical space ($x$, $y$). Trials consisted of the following events: (1) a black loading screen for 500 ms, (2) stimulus presentation for 500 ms, (3) a black screen for 600 ms, (4) a response recording window which continued until a response was made, and (5) trialwise feedback for 800 ms. When space was the pre-training modality, the response was made by clicking on 1 of 16 spatial locations on a 4×4 grid. When vision was the pre-training modality, response was made by clicking on 1 of 16 visual shapes arranged on a 4×4 grid. The spatial arrangement of the visual shapes changed randomly every block (48 trials) to deconfound spatial and visual features. The ordering of the trials was pseudo-randomised such that each of the 16 stimuli appeared three times each every block (48 trials). On top of trialwise feedback, participants received blockwise feedback on their performance in the last block. Finally, the mapping task could be restricted to a given dimension while fixing the other dimension, e.g., only change in the $i$ dimension while maintaining the $j$ dimension at a constant value.

## Filler task (Exp. 4 only)

In Exp. 4, a duration-matched filler task was introduced to replace the pre-training task, ensuring that the number of trials was kept constant and removing any exposure to the categorisation task in the spatial modality (see *Figure 1—figure supplement 3C*). As in the main task, participants were asked to infer the category of sequences of four items. There were three possible categories. The sequences were composed of four coloured stars appearing at the same location in space: either red-red-red-blue, red-red-blue-blue, or red-blue-blue-blue. Trials consisted of the following events: (1) a black loading screen for 500 ms, (2) sequence presentation for 8000 ms (four times 500 ms of stimulus presentation followed by 1500 ms black screen), (3) a response recording window which continued until a response was made, and (4) trialwise feedback for 800 ms. Response was made by clicking with the mouse on one of three buttons that appeared on screen. The ordering of the buttons was randomised across participants, and kept fixed for the entire experiment. The buttons were different from those used in the main task. The ordering of the trials was pseudo-randomised such that the three sequence categories appeared 10 times each every block (30 trials). The location of the star was chosen randomly every trial among the 16 possible locations. Participants were instructed on the

**Table 2.** Procedure for Experiment 1.

The table reads as follows: on day 1, participants in the Exp. 1a began with 120 trials of the quadruplet categorisation task in the auditory modality, with canonical quadruplets (0°, denoted by ∅), with feedback on every trials. They subsequently performed 105 trials (with trialwise feedback) and 105 transfer trials including rotated and far transfer quadruplets (without trialwise feedback) which were presented in mixed blocks of 30 trials. Training and transfer trials were randomly interleaved, and no clue indicated whether participants were currently on a training trial or a transfer trial before feedback (or absence of feedback in case of a transfer trial). All participants thus performed a total of 330 trials in a single session. Abbreviations/symbols: Fb.: trialwise feedback. Trans.: transformations of the quadruplets. Transformations are canonical (∅), 90° rotation (↷), 180° rotation (↻), 270° rotation (↶), far transfer canonical (★), far transfer 90° rotation (★ + ↷), far transfer 180° rotation (★ + ↻), and far transfer 270° rotation (★ + ↶).

| Day | Task | Trial # | Trans. | Fb. | Exp. 1a (*N*=50) | Exp. 1b (*N*=52) | Exp. 1c (*N*=51) |
|---|---|---|---|---|---|---|---|
| | Quadruplet categorisation | 120 | ∅ | Yes | Auditory | Visual | Spatial |
| | | 105 | ∅ | Yes | Auditory | Visual | Spatial |
| 1 | Quadruplet categorisation | 105 | ↷<br>↻<br>↶<br>★<br>★ + ↷<br>★ + ↻<br>★ + ↶ | No | Auditory | Visual | Spatial |

deterministic nature of the task. ('*The rules used by the aliens to produce the sequences are 100% deterministic. This means that once you have discovered the rules, you will reach 100% of correct responses*'.) On top of trialwise feedback, participants received blockwise feedback on their performance in the last block.

## Multi-day experiments

Exp. 2, 3, and 4 took place over the course of 2 days. After having completed the 'day 1' of the experiment, participants were proposed the 'day 2' of the experiment after 24 hr. If no completion of day 2 had been received after 72 hr, participants were considered dropped out.

## Complete task schedule

The ordering of the tasks and their characteristics varied across experiments. The following tables summarise the task schedules for Exp. 1 (see *Table 2*), Exp. 2 (see *Table 3*), Exp. 3 (see *Table 4*), and Exp. 4 (see *Table 5*).

## **Statistical analysis**

### Outliers

No outliers were removed from the analyses.

### Inference models

We designed inference models that used different kinds of representation to make an inference about the quadruplet category. These models were fit to each participant's choices in order to decipher the most likely strategy they were using during training, near transfer, and far transfer.

There were seven models to fit the near transfer data (see *Figure 1—figure supplement 4*):

- *R*: a random model that responds randomly to every trial (null model).
- *R'*: another random model that responds correctly to the training trials but randomly to the transfer trials ('non-generaliser' or 'overfitting' model).
- *1Di_u*: a model that responds according to the unsigned transitions in the *i* dimension, such as 'ABAB', 'ABBA', and 'AABB' (where A and B are two feature locations on the *i* dimension). As the model responds in an unsigned manner, 'ABAB' maps onto 'BABA', 'ABBA' onto 'BAAB', and 'AABB' onto 'BBAA'. This model achieves 100% accuracy in the training trials in Exp. 1 but

**Table 3.** Procedure for Experiment 2.

Same as *Table 2*. All participants thus performed a total of 396 trials on day 1 and 600 trials on day 2. Sp. → Aud.: spatial to auditory mapping task. Sp. → Vis.: spatial to visual mapping task. Vis. → Aud.: visual to auditory mapping task.

| Day | Task | Trial # | Trans. | Fb. | Exp. 2a (*N*=51) | Exp. 2b (*N*=51) | Exp. 2c (*N*=50) |
|---|---|---|---|---|---|---|---|
| | Quadruplet categorisation | 60 | ∅ ↻ | Yes | Spatial | Spatial | Visual |
| | | 48 | *i* | Yes | Sp. → Aud. | Sp. → Vis. | Vis. → Aud. |
| | | 48 | *j* | Yes | Sp. → Aud | Sp. → Vis. | Vis. → Aud. |
| | Mapping | 48 | *ij* | Yes | Sp. → Aud | Sp. → Vis. | Vis. → Aud. |
| | | 48 | *i* | Yes | Sp. → Aud | Sp. → Vis. | Vis. → Aud. |
| | | 48 | *j* | Yes | Sp. → Aud | Sp. → Vis. | Vis. → Aud. |
| 1 | Mapping | 96 | *ij* | Yes | Sp. → Aud | Sp. → Vis. | Vis. → Aud. |
| | Quadruplet categorisation | 60 | ∅ ↻ | Yes | Spatial | Spatial | Visual |
| | | 48 | *i* | Yes | Sp. → Aud | Sp. → Vis. | Vis. → Aud. |
| | | 48 | *j* | Yes | Sp. → Aud | Sp. → Vis. | Vis. → Aud. |
| | Mapping | 48 | *ij* | Yes | Sp. → Aud | Sp. → Vis. | Vis. → Aud. |
| | | 30 | ∅ ↻ | Yes | Sp.+Aud | Sp.+Vis. | Vis.+Aud |
| | Quadruplet categorisation | 30 | ∅ ↻ | Yes | Auditory | Visual | Auditory |
| | Mapping | 48 | *ij* | Yes | Sp. → Aud | Sp. → Vis. | Vis. → Aud |
| | Quadruplet categorisation | 60 | ∅ ↻ | Yes | Auditory | Visual | Auditory |
| | Mapping | 48 | *ij* | Yes | Sp. → Aud | Sp. → Vis. | Vis. → Aud |
| | | 90 | ∅ ↻ | Yes | Auditory | Visual | Auditory |
| 2 | Quadruplet categorisation | 90 | ↺ ↻ ★ ★ + ↻ ★ + ↺ ★ + ↻ | No | Auditory | Visual | Auditory |

50% accuracy in the training trials in Exp. 2, 3, and 4. This is because when both canonical (0°) and 90° rotated quadruplets are present, the unsigned transitions in either dimension are not fully diagnostic of the category. For example, the pattern 'ABBA' in the *j* dimension correspond to both the category 0 with 0° rotation and category 1 with 90° rotation (see *Figure 1—figure supplement 2*).

- *1Dj$_u$*: same as *1Di$_u$* but in the *i* dimension.
- *1Di$_s$*: a model that responds to the signed transitions in the *i* dimension, such as 'ABAB', 'BABA', 'ABBA', 'BAAB', 'AABB', and 'BBAA' (where A and B are two feature locations on the *i* dimension, and A is lower than B). As the model responds in a signed manner, 'ABAB' does not map onto 'BABA'. This model achieves 100% accuracy in the training trials in Exp. 1, 2, 3, and 4.
- *1Dj$_s$*: same as *1Di$_s$* but in the *j* dimension. This model achieves 100% accuracy in the training trials in Exp. 1 but 50% accuracy in Exp. 2, 3, and 4. This is again because when both canonical (0°) and 90° rotated quadruplets are present, the signed transitions in the *j* dimension are not 100% diagnostic of the quadruplet category.
- *2D*: a model that responds according to the vector transitions in both *i* and *j* dimensions. This model trivially achieves 100% accuracy in the training trials in Exp. 1, 2, 3, and 4.

**Table 4.** Procedure for Experiment 3.
Same conventions as *Table 2*.

| Day | Task | Trial # | Trans. | Fb. | Exp. 3a (*N*=50) | Exp. 3b (*N*=51) | Exp. 3c (*N*=50) |
|---|---|---|---|---|---|---|---|
| | Quadruplet categorisation | 60 | ∅ | Yes | Spatial | Spatial | Visual |
| | | 48 | *i* | Yes | Sp. → Aud. | Sp. → Vis. | Vis. → Aud. |
| | | 48 | *j* | Yes | Sp. → Aud | Sp. → Vis. | Vis. → Aud. |
| | Mapping | 48 | *ij* | Yes | Sp. → Aud | Sp. → Vis. | Vis. → Aud. |
| | | 48 | *i* | Yes | Sp. → Aud | Sp. → Vis. | Vis. → Aud. |
| | | 48 | *j* | Yes | Sp. → Aud | Sp. → Vis. | Vis. → Aud. |
| 1 | Mapping | 96 | *ij* | Yes | Sp. → Aud | Sp. → Vis. | Vis. → Aud. |
| | Quadruplet categorisation | 60 | ∅ | Yes | Spatial | Spatial | Visual |
| | | 48 | *i* | Yes | Sp. → Aud | Sp. → Vis. | Vis. → Aud. |
| | | 48 | *j* | Yes | Sp. → Aud | Sp. → Vis. | Vis. → Aud. |
| | Mapping | 48 | *ij* | Yes | Sp. → Aud | Sp. → Vis. | Vis. → Aud. |
| | | 30 | ∅ | Yes | Sp.+Aud | Sp.+Vis. | Vis.+Aud |
| | Quadruplet categorisation | 30 | ∅ ↶ | Yes | Auditory | Visual | Auditory |
| | Mapping | 48 | *ij* | Yes | Sp. → Aud | Sp. → Vis. | Vis. → Aud |
| | Quadruplet categorisation | 60 | ∅ ↶ | Yes | Auditory | Visual | Auditory |
| | Mapping | 48 | *ij* | Yes | Sp. → Aud | Sp. → Vis. | Vis. → Aud |
| | | 90 | ∅ ↶ | Yes | Auditory | Visual | Auditory |
| 2 | Quadruplet categorisation | 90 | ↻ ↶ ★ ★ + ↶ ★ + ↻ ★ + ↶ | No | Auditory | Visual | Auditory |

Four more models were added when fitting the far transfer data to account for the fact that the participant can map between dimensions in the original manifold and dimensions in the far transfer manifold in a variety of ways (see *Figure 1—figure supplement 5*). For example, a participant tracking patterns in the *i* dimension during training could track the same pattern in the ★*j* dimension in far transfer.

- *1Dij$_u$*: a model that tracks the unsigned transitions in the *i* dimension and respond as if ★*j* was the *i* dimension in far transfer.
- *1Dji$_u$*.
- *1Dij$_s$*.
- *1Dji$_s$*.

## Model likelihood

All models, except the random model *R*, had one free parameter: the temperature parameter $\beta$ of a softmax when converting inference over category into choice probability. For a single trial, the likelihood was defined as:

**Table 5.** Procedure for Experiment 4.
Same as *Table 2*.

| Day | Task | Trial # | Trans. | Fb. | Exp. 4a (N=50) | Exp. 4b (N=52) |
|---|---|---|---|---|---|---|
| | Filler | 60 | | Yes | Spatial | Spatial |
| | | 48 | i | Yes | Sp. → Aud. | Sp. → Vis. |
| | | 48 | j | Yes | Sp. → Aud | Sp. → Vis. |
| | Mapping | 48 | ij | Yes | Sp. → Aud | Sp. → Vis. |
| | | 48 | i | Yes | Sp. → Aud | Sp. → Vis. |
| | | 48 | j | Yes | Sp. → Aud | Sp. → Vis. |
| 1 | Mapping | 96 | ij | Yes | Sp. → Aud | Sp. → Vis. |
| | Filler | 60 | Ø | Yes | Spatial | Spatial |
| | | 48 | i | Yes | Sp. → Aud | Sp. → Vis. |
| | | 48 | j | Yes | Sp. → Aud | Sp. → Vis. |
| | Mapping | 48 | ij | Yes | Sp. → Aud | Sp. → Vis. |
| | Quadruplet categorisation | 60 | Ø ↶ | Yes | Auditory | Visual |
| | Mapping | 48 | ij | Yes | Sp. → Aud | Sp. → Vis. |
| | Quadruplet categorisation | 60 | Ø ↶ | Yes | Auditory | Visual |
| | Mapping | 48 | ij | Yes | Sp. → Aud | Sp. → Vis. |
| | | 90 | Ø ↶ | Yes | Auditory | Visual |
| 2 | Quadruplet categorisation | 90 | ↺ ↶ ★ ★ + ↶ ★ + ↺ ★ + ↶ | No | Auditory | Visual |

$$\mathcal{L}(C_{p,t}, Q_{p,t}, \mathcal{M}, \beta) = \frac{e^{\frac{\mathrm{p}(C_{p,t}|Q_{p,t}, \mathcal{M})}{\beta}}}{\sum_{c=0}^{2} e^{\frac{\mathrm{p}(c|Q_{p,t}, \mathcal{M})}{\beta}}}$$

where $C_{p,t}$ is the category chosen by the participant $p$ on trial $t$ ($C_{p,t}$=0, 1, or 2), $Q_{p,t}$ the quadruplet presented on this trial, $M$ the inference model, $\beta$ the temperature parameter, and $\mathrm{p}(c|Q_{p,t}, M)$ the probability assigned by model $M$ to the category $c$ for the quadruplet $Q_{p,t}$.

Assuming that trials are independent, the likelihood of model $M$ for participant $p$ over all trials is the product of the likelihood of the individual trials, or equivalently, the log-likelihood is the sum of the log-likelihood of the individual trials:

$$\ln(\mathcal{L}_p(\mathcal{M}, \beta)) = \sum_{t=0}^{T-1} \ln(\mathcal{L}(C_{p,t}, Q_{p,t}, \mathcal{M}, \beta))$$

## Model fitting

For models with a temperature parameter $\beta$, the maximum likelihood was defined as the maximum value of the likelihood function over 200 linearly spaced values of $\beta$ between 0.01 and 0.5.

$$\hat{\mathcal{L}}_p(\mathcal{M}) = \max_{\beta}(\mathcal{L}_p(\mathcal{M}, \beta))$$

**Table 6.** Bayes factors quantifying the support in favour of a difference in model frequencies between each pair of conditions in the near transfer condition: *1D* vs *2D*.

Models were grouped in two families as follows: [$1Di_u$, $1Dj_u$, $1Di_s$, $1Dj_s$], [2D]. A BF>1 indicates evidence in favour of a difference in model frequencies (green, BF>3, BF>10, and BF>100 corresponds respectively to substantial, strong, and decisive evidence in favour of a difference in model frequencies between groups). A BF<1 indicates evidence in favour of similar model frequencies (red, BF<0.3, BF<0.1, and BF<0.01 corresponds respectively to substantial, strong, and decisive evidence in favour of no difference in model frequencies between groups).

| | Exp. 1b | Exp. 1c | Exp. 2a | Exp. 2b | Exp. 2c | Exp. 3a | Exp. 3b | Exp. 3c | Exp. 4a | Exp. 4b |
|---|---|---|---|---|---|---|---|---|---|---|
| Exp. 1a | 0.1 | $1.1\times10^{17}$ | $4.5\times10^{10}$ | $1.0\times10^{15}$ | 0.9 | $8.1\times10^4$ | $6.6\times10^7$ | 0.2 | 85.8 | 730.1 |
| Exp. 1b | | $3.8\times10^7$ | $1.1\times10^{11}$ | $3.1\times10^{15}$ | 1.0 | $1.2\times10^5$ | $1.3\times10^8$ | 0.2 | 106.7 | 979.4 |
| Exp. 1c | | | 2.0 | 0.2 | $6.3\times10^{12}$ | 6999.7 | 40.3 | $2.1\times10^{15}$ | $9.6\times10^7$ | $4.2\times10^6$ |
| Exp. 2a | | | | 0.6 | $9.0\times10^6$ | 3.4 | 0.3 | $1.2\times10^9$ | 1630.6 | 169.8 |
| Exp. 2b | | | | | $8.3\times10^{10}$ | 483.7 | 5.9 | $2.2\times10^{13}$ | $2.6\times10^6$ | $1.4\times10^5$ |
| Exp. 2c | | | | | | 122.5 | $3.1\times10^4$ | 0.3 | 0.9 | 3.6 |
| Exp. 3a | | | | | | | 0.4 | 4220.0 | 0.9 | 0.3 |
| Exp. 3b | | | | | | | | $2.4\times10^6$ | 26.6 | 4.8 |
| Exp. 3c | | | | | | | | | 8.6 | 56.0 |
| Exp. 4a | | | | | | | | | | 0.2 |

For each participant, the best model was chosen as the model with the lowest Bayesian information criterion (BIC). This was done to adjust for model complexity between models without parameters (the random model *R*) and models with one parameter (all the others). For each participant *p* and model *M*, $BIC_p(M)$ was defined as:

$$BIC_P(\mathcal{M}) = k\ln(T) - 2\ln(\hat{\mathcal{L}}_p(\mathcal{M}))$$

$$BIC_P(\mathcal{M}) = k\ln(T) - 2\ln(\hat{\mathcal{L}}_p(\mathcal{M}))$$

where *k* is the number of parameters (k=0 for the random model *R*, k=1 for all other models) and *T* the number of trials.

The inference models were fitted to trial-by-trial choice data independently for each participant using training and near transfer trials for near transfer and using training and far transfer trials for far

**Table 7.** Bayes factors quantifying the support in favour of a difference in model frequencies between each pair of conditions in the near transfer condition: *R* vs *1D/2D*.

Models were grouped in two families as follows: [*R*, *R'*], [$1Di_u$, $1Dj_u$, $1Di_s$, $1Dj_s$, 2D].

| | Exp. 1b | Exp. 1c | Exp. 2a | Exp. 2b | Exp. 2c | Exp. 3a | Exp. 3b | Exp. 3c | Exp. 4a | Exp. 4b |
|---|---|---|---|---|---|---|---|---|---|---|
| Exp. 1a | 1.7 | 17.7 | 2.7 | 9.7 | 7.4 | 0.3 | 0.2 | 0.2 | 0.2 | 0.3 |
| Exp. 1b | | 0.2 | 0.2 | 0.2 | $2.1\times10^5$ | 0.3 | 0.4 | 0.8 | 0.5 | 20.4 |
| Exp. 1c | | | 0.2 | 0.2 | $1.4\times10^6$ | 1.0 | 2.3 | 6.1 | 2.7 | 437.0 |
| Exp. 2a | | | | 0.2 | $4.0\times10^5$ | 0.3 | 0.6 | 1.2 | 0.7 | 36.0 |
| Exp. 2b | | | | | $4.6\times10^6$ | 0.7 | 1.5 | 3.6 | 1.7 | 196.9 |
| Exp. 2c | | | | | | 198.0 | 47.7 | 13.1 | 34.0 | 0.8 |
| Exp. 3a | | | | | | | 0.2 | 0.2 | 0.2 | 1.3 |
| Exp. 3b | | | | | | | | 0.2 | 0.2 | 0.6 |
| Exp. 3c | | | | | | | | | 0.2 | 0.3 |
| Exp. 4a | | | | | | | | | | 0.5 |

**Table 8.** Bayes factors quantifying the support in favour of a difference in model frequencies between each pair of conditions in the far transfer condition: *1D* vs *2D*.

Models were grouped in two families as follows: [*1Di*$_u$, *1Dij*$_u$, *1Dj*$_u$, *1Dji*$_u$ *1Di*$_s$, *1Dij*$_s$, *1Dj*$_s$, *1Dji*$_s$], [*2D*].

| | Exp. 1b | Exp. 1c | Exp. 2a | Exp. 2b | Exp. 2c | Exp. 3a | Exp. 3b | Exp. 3c | Exp. 4a | Exp. 4b |
|---|---|---|---|---|---|---|---|---|---|---|
| Exp. 1a | 0.1 | $2.0\times10^{12}$ | $4.0\times10^{10}$ | $3.5\times10^{11}$ | 1.9 | $3.4\times10^{5}$ | $8.5\times10^{5}$ | 0.4 | 88.7 | 366.1 |
| Exp. 1b | | $4.6\times10^{12}$ | $8.4\times10^{10}$ | $7.6\times10^{11}$ | 2.1 | $5.1\times10^{5}$ | $1.3\times10^{6}$ | 0.4 | 107.2 | 461.3 |
| Exp. 1c | | | 0.2 | 0.2 | $3.6\times10^{7}$ | 7.2 | 4.1 | $2.8\times10^{9}$ | $2.3\times10^{4}$ | 3403.6 |
| Exp. 2a | | | | 0.2 | $1.2\times10^{6}$ | 1.5 | 1.0 | $7.3\times10^{7}$ | 1439.1 | 266.4 |
| Exp. 2b | | | | | $8.0\times10^{6}$ | 3.7 | 2.2 | $5.6\times10^{8}$ | 6820.3 | 1126.7 |
| Exp. 2c | | | | | | 101.4 | 204.2 | 0.2 | 0.5 | 1.0 |
| Exp. 3a | | | | | | | 0.2 | 1976.8 | 1.6 | 0.7 |
| Exp. 3b | | | | | | | | 4445.1 | 2.6 | 1.0 |
| Exp. 3c | | | | | | | | | 2.4 | 6.8 |
| Exp. 4a | | | | | | | | | | 0.2 |

transfer. Using training trials was done to improve the fits, as some models differ in their response during training, e.g., model *1Di*$_u$ and *1Dj*$_u$ in Exp. 2, 3, and 4.

## Model recovery

A model recovery analysis was performed to ensure that the experimental design was able to differentiate between models. We generated artificial data for each model with the same trials and the same number of trials as our human participants. We simulated 100 models for four values of the temperature parameter (0.05, 0.2, 0.35, and 0.5). Results showed that model recovery was very good for all experiments, even in high noise regimes (temperature of 0.5) (see *Figure 1—figure supplement 6*).

## Model comparison

Model frequencies and difference in model frequencies between groups were estimated using Bayesian group comparison as described in *Rigoux et al., 2014*. The marginal likelihood for model *M* and choice data $C_p$ of participant *p* was estimated using *BIC* and defined as:

**Table 9.** Bayes factors quantifying the support in favour of a difference in model frequencies between each pair of conditions in the far transfer condition: *R* vs *1D/2D*.

Models were grouped in two families as follows: [*R*, *R'*], [*1Di*$_u$, *1Dij*$_u$, *1Dj*$_u$, *1Dji*$_u$, *1Di*$_s$, *1Dij*$_s$, *1Dj*$_s$, *1Dji*$_s$, *2D*].

| | Exp. 1b | Exp. 1c | Exp. 2a | Exp. 2b | Exp. 2c | Exp. 3a | Exp. 3b | Exp. 3c | Exp. 4a | Exp. 4b |
|---|---|---|---|---|---|---|---|---|---|---|
| Exp. 1a | 0.5 | 0.2 | 1.2 | 1.4 | 2.7 | 0.3 | 0.6 | 0.2 | 0.2 | 9.6 |
| Exp. 1b | | 0.2 | 0.2 | 131.0 | 340.4 | 0.2 | 22.9 | 0.4 | 0.3 | 2635.3 |
| Exp. 1c | | | 0.3 | 14.5 | 33.4 | 0.2 | 3.6 | 0.2 | 0.2 | 185.7 |
| Exp. 2a | | | | 525.3 | 1412.4 | 0.3 | 81.1 | 0.9 | 0.6 | $1.2\times10^{4}$ |
| Exp. 2b | | | | | 0.2 | 14.1 | 0.2 | 1.5 | 3.0 | 0.2 |
| Exp. 2c | | | | | | 31.8 | 0.2 | 2.8 | 6.2 | 0.2 |
| Exp. 3a | | | | | | | 3.6 | 0.2 | 0.2 | 167.7 |
| Exp. 3b | | | | | | | | 0.6 | 1.0 | 0.4 |
| Exp. 3c | | | | | | | | | 0.2 | 9.9 |
| Exp. 4a | | | | | | | | | | 25.7 |

$$p(C_p|\mathcal{M}) \approx e^{-\frac{1}{2}BIC_p(\mathcal{M})}$$

This estimate was used to compute the posterior probability $p(H_0|C)$, which quantifies the probability that two groups come from the same distribution, i.e., have similar model frequencies. Under uniform prior over $H_0$ and $H_1$ (the two groups do not come from the same distribution), this allowed to compute a *BF* as follows:

$$BF = \frac{p(C|H_1)}{p(C|H_0)} = \frac{p(H_1|C)}{p(H_0|C)}\frac{p(H_0)}{p(H_1)} = \frac{1p(H_0|C)}{p(H_0|C)}$$

In this form, the BF quantifies the support of the data in favour of a difference in model frequencies between groups. We followed *Kass and Raftery, 1995* for the interpretation of its values: BF>3, BF>10, and BF>100 were respectively taken as substantial, strong, and decisive evidence in favour of a difference in model frequencies between groups (BF<0.3, BF<0.1, and BF<0.01 as evidence in favour of no difference in model frequencies).

## Cross-validation visualisation
Finally, cross-validation was used for visualisation. For this, we first fitted the models using half of the trials (even trial numbers) and selected the model with the lowest BIC for each participant. We then computed the response matrix of each participant using the unobserved half of the trials (odd trial numbers). We finally displayed the averaged left-one-out response matrices and the expected response matrix for models that had been selected as the best model for at least five participants.

## Acknowledgements

We thank Jean Daunizeau for technical help with modelling. Funding Work supported by Fondation Pour l'Audition FPA RD-2021-2 (JPL) and European Research Council Consolidator Grant n° 725937 - CQR01290.CQ001 (CS).

## Additional information

### Funding

| Funder | Grant reference number | Author |
| --- | --- | --- |
| Fondation Pour l'Audition | FPA RD-2021-2 | Jacques Pesnot Lerousseau |
| European Research Council | 725937 - CQR01290.CQ001 | Christopher Summerfield |

The funders had no role in study design, data collection and interpretation, or the decision to submit the work for publication.

### Author contributions
Jacques Pesnot Lerousseau, Conceptualization, Data curation, Formal analysis, Funding acquisition, Investigation, Visualization, Methodology, Writing - original draft; Christopher Summerfield, Conceptualization, Resources, Supervision, Funding acquisition, Methodology, Writing - original draft, Project administration, Writing - review and editing

### Author ORCIDs
Jacques Pesnot Lerousseau ⓘ http://orcid.org/0000-0003-3799-0602

### Ethics
All experiments were approved by the Medical Sciences Research Ethics Committee of the University of Oxford (approval reference R50750/RE005). Before starting the experiment, informed consent was taken through an online form, and participants indicated that they understood the goals of the study,

how to raise any questions, how their data would be handled, and that they were free to withdraw from the experiment at any time.

Reviewer #1 (Public Review): https://doi.org/10.7554/eLife.93636.3.sa1
Reviewer #2 (Public Review): https://doi.org/10.7554/eLife.93636.3.sa2
Author Response https://doi.org/10.7554/eLife.93636.3.sa3

---

# Additional files

## Supplementary files
• MDAR checklist

## Data availability
Anonymized data, code, materials and pre-registration documents are all available at https://osf.io/z9572/.

The following dataset was generated:

| Author(s) | Year | Dataset title | Dataset URL | Database and Identifier |
|---|---|---|---|---|
| Pesnot Lerousseau J, Summerfield C | 2023 | Space as a Scaffold for Rotational Generalisation of Abstract Concepts | https://osf.io/z9572/ | Open Science Framework, 10.17605/OSF.IO/Z9572 |

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
