## [Editor Report · eLife assessment]

These ingenious and thoughtful studies present **important** findings concerning how people can represent and generalise abstract patterns of sensory data. The issue of generalization is a core topic in neuroscience and psychology, relevant across a wide range of areas, and the findings will be of interest to researchers across areas in perception, learning and cognitive science. The findings are **convincing** in this setting, but future research must establish their generality and interrogate the precise nature of the underlying mechanism.

---

## [Referee Report · Reviewer #1 (Public Review)]

Summary:

This manuscript reports a series of experiments examining category learning and subsequent generalization of stimulus representations across spatial and nonspatial domains. In Experiment 1, participants were first trained to make category judgments about sequences of stimuli presented either in nonspatial auditory or visual modalities (with feature values drawn from a two-dimensional feature manifold, e.g., pitch vs timbre), or in a spatial modality (with feature values defined by positions in physical space, e.g., Cartesian x and y coordinates). A subsequent test phase assessed category judgments for 'rotated' exemplars of these stimuli: i.e., versions in which the transition vectors are rotated in the same feature space used during training (near transfer) or in a different feature space belonging to the same domain (far transfer). Findings demonstrate clearly that representations developed for the spatial domain allow for representational generalization, whereas this pattern is not observed for the nonspatial domains that are tested. Subsequent experiments demonstrate that if participants are first pre-trained to map nonspatial auditory/visual features to spatial locations, then rotational generalization is facilitated even for these nonspatial domains. It is argued that these findings are consistent with the idea that spatial representations form a generalized substrate for cognition: that space can act as a scaffold for learning abstract nonspatial concepts.

Strengths:

I enjoyed reading this manuscript, which is extremely well written and well presented. The writing is clear and concise throughout, and the figures do a great job of highlighting the key concepts. The issue of generalization is a core topic in neuroscience and psychology, relevant across a wide range of areas, and the findings will be of interest to researchers across areas in perception and cognitive science. It's also excellent to see that the hypotheses, methods and analyses were pre-registered.

The experiments that have been run are ingenious and thoughtful; I particularly liked the use of stimulus structures that allow for disentangling of one-dimensional and two-dimensional response patterns. The studies are also well powered for detecting effects of interest. The model-based statistical analyses are thorough and appropriate throughout (and it's good to see model recovery analysis too). The findings themselves are clear-cut: I have little doubt about the robustness and replicability of these data.

Weaknesses:

In my original review I raised a concern related to a potential alternative interpretation of the findings: the idea that participants have substantial experience of representing space in terms of multiple, independent, and separable dimensions, whereas this may not be the case for the visual and auditory stimuli used here. As I noted in that prior review, on this view "the impact of spatial pre-training and (particularly) mapping is simply to highlight to participants that the auditory / visual stimuli comprise two separable (and independent) dimensions."

In addressing this point, the authors note that performance in the visual/auditory "mapping" task in Experiments 2c and 3c suggests that most participants were paying attention to both dimensions of auditory and visual stimuli. I agree that seems to have been the case. But there is a difference between making use of information from both dimensions, and realizing that ***the two dimensions are separable and independent*** (which is what is required for rotational generalization in this task).

As an analogy, suppose I have a task where participants have to map a pillow and a shuttlecock to category A, and a surfboard and a bicycle to category B. A participant could learn to do this just by memorizing the correct response for each item considered as a "whole thing". Or they could realize that the items contain component information, learning that "things with feathers" belong in category A, and "things that can carry people" go in category B. Performance may be the same in both cases, but the underlying process is quite different.

The "attention to dimensions" account that I advanced in my previous review was referring to something more like the latter (feathers/vehicle) case: that spatial pre-training helps people to understand that items can be decomposed into separable pieces of information. Above-chance performance in the visual-auditory mapping task does not (necessarily) demonstrate this ability because it could reflect memorization of "whole" stimuli rather than reflecting decomposition into separable component parts. I agree that it does at least show that participants were paying attention to and making use of information from both dimensions when making their mapping decisions; it's just that they may not have *realized* that they were using information from two separable dimensions.

---

## [Referee Report · Reviewer #2 (Public Review)]

Summary:

In this manuscript, L&S investigates the important general question of how humans achieve invariant behavior over stimuli belonging to one category given the widely varying input representation of those stimuli and more specifically, how they do that in arbitrary abstract domains. The authors start with the hypothesis that this is achieved by invariance transformations that observers use for interpreting different entries and furthermore, that these transformations in an arbitrary domain emerge with the help of the transformations (e. g. translation, rotation) within the spatial domain by using those as "scaffolding" during transformation learning. To provide the missing evidence for this hypothesis, L&S used behavioral category learning studies within and across the spatial, auditory and visual domains, where rotated and translated 4-element token sequences had to be learned to categorize and then the learned transformation had to applied in new feature dimensions within the given domain. Through single- and multiple-day supervised training and unsupervised tests, L&S demonstrated by standard computational analyses that in such setups, space and spatial transformations can, indeed, help with developing and using appropriate rotational mapping whereas the visual domain cannot fulfill such a scaffolding role.

Strengths:

The overall problem definition and the context of spatial mapping-driven solution to the problem is timely. The general design of testing the scaffolding effect across different domains is more advanced than any previous attempts clarifying the relevance of spatial coding to any other type of representational codes. Once the formulation of the general problem in a specific scientific framework is done, the following steps are clearly and logically defined and executed. The obtained results are well interpretable, and they could serve as a good steppingstone for deeper investigations. The analytical tools used for the interpretations are adequate. The paper is relatively clearly written.

Weaknesses:

Some additional effort to clarify the exact contribution of the paper, the link between analyses and the claims of the paper and its link to previous proposals would be necessary to better assess the significance of the results and the true nature of the proposed mechanism of abstract generalization.

(1) Insufficient conceptual setup: The original theoretical proposal (the Tolman-Eichenbaum-Machine, Whittington et al., Cell 2020) that L&S relate their work proposes that just as in the case of memory for spatial navigation, humans and animal create their flexible relational memory system of any abstract representation by a conjunction code that combines on the one hand, sensory representation and on the other hand, a general structural representation or relational transformation. The TEM also suggest that the structural representation could contain any graph-interpretable spatial relations, albeit in their demonstration 2D neighbor relations were used. The goal of L&S's paper is to provide behavioral evidence for this suggestion by showing that humans use representational codes that are invariant to relational transformations of non-spatial abstract stimuli and moreover, that humans obtain these invariances by developing invariance transformers with the help of available spatial transformers. To obtain such evidence, L&S use the rotational transformation. However, the actual procedure they used actually solved an alternative task: instead of interrogating how humans develop generalizations in abstract spaces, they demonstrated that if one defines rotation in an abstract feature space embedded in visual or auditory modality that is similar to the 2D space (i.e. has two independent dimensions that are clearly segregable and continuous), humans cannot learn to apply rotation of 4-piece temporal sequences in those spaces while they can do it in 2D space, and with co-associating a one-to-one mapping between locations in those feature spaces with locations in the 2D space an appropriate shaping mapping training will lead to successful application of rotation in the given task (and in some other feature spaces in the given domain). While this is an interesting and challenging demonstration, it does not shed light on how humans learn and generalize only that humans CAN do learning and generalization in this, highly constrained scenario. This result is a demonstration of how a stepwise learning regiment can make use of one structure for mapping a complex input into a desired output. The results neither clarify how generalizations would develop in abstract spaces nor the question if this generalization uses transformations developed in the abstract space. The specific training procedure ensures success in the presented experiments but the availability and feasibility of an equivalent procedure in natural setting is a crucial part of validating the original claim and that has not been done in the paper.

(2) Missing controls: The asymptotic performance in Exp 1 after training in the three tasks was quite different in the three tasks (intercepts 2.9, 1.9, 1.6 for spatial, visual and auditory, respectively; p. 5. para. 1, Fig 2BFJ). It seems that the statement "However, or main question was how participants would generalise learning to novel, rotated exemplars of the same concept." assumes that learning and generalization are independent. Wouldn't it be possible, though, that the level of generalization depends on the level of acquiring a good representation of the "concept" and after obtaining an adequate level of this knowledge, generalization would kick in without scaffolding? If so, a missing control is to equate the levels of asymptotic learning and see whether there is a significant difference in generalization. A related issue is that we have no information what kind of learning in the three different domains were performed, albeit we probably suspect that in space the 2D representation was dominant while in the auditory and visual domains not so much. Thus, a second missing piece of evidence is the model fitting results of the ⦰ condition that would show which way the original sequences were encoded (similar to Fig 2 CGK and DHL). If the reason for lower performance is not individual stimulus difficulty but the natural tendency to encode the given stimulus type by a combo of random + 1D strategy that would clarify that the result of the cross-training is, indeed, transferring the 2D-mapping strategy.

---

## [Author Response]

The following is the authors’ response to the original reviews.

**eLife assessment**
These ingenious and thoughtful studies present important findings concerning how people represent and generalise abstract patterns of sensory data. The issue of generalisation is a core topic in neuroscience and psychology, relevant across a wide range of areas, and the findings will be of interest to researchers across areas in perception, learning, and cognitive science. The findings have the potential to provide compelling support for the outlined account, but there appear other possible explanations, too, that may affect the scope of the findings but could be considered in a revision.

Thank you for sending the feedback from the three peer reviewers regarding our paper. Please find below our detailed responses addressing the reviewers' comments. We have incorporated these suggestions into the paper and provided explanations for the modifications made.

We have specifically addressed the point of uncertainty highlighted in eLife's editorial assessment, which concerned alternative explanations for the reported effect. In response to Reviewer #1, we have clarified how Exp. 2c and Exp. 3c address the potential alternative explanation related to "attention to dimensions." Further, we present a supplementary analysis to account for differences in asymptotic learning, as noted by Reviewer #2. We have also clarified how our control experiments address effects associated with general cognitive engagement in the task. Lastly, we have further clarified the conceptual foundation of our paper, addressing concerns raised by Reviewers #2 and #3.

**Reviewer #1 (Public Review):**
Summary:This manuscript reports a series of experiments examining category learning and subsequent generalization of stimulus representations across spatial and nonspatial domains. In Experiment 1, participants were first trained to make category judgments about sequences of stimuli presented either in nonspatial auditory or visual modalities (with feature values drawn from a two-dimensional feature manifold, e.g., pitch vs timbre), or in a spatial modality (with feature values defined by positions in physical space, e.g., Cartesian x and y coordinates). A subsequent test phase assessed category judgments for 'rotated' exemplars of these stimuli: i.e., versions in which the transition vectors are rotated in the same feature space used during training (near transfer) or in a different feature space belonging to the same domain (far transfer). Findings demonstrate clearly that representations developed for the spatial domain allow for representational generalization, whereas this pattern is not observed for the nonspatial domains that are tested. Subsequent experiments demonstrate that if participants are first pre-trained to map nonspatial auditory/visual features to spatial locations, then rotational generalization is facilitated even for these nonspatial domains. It is argued that these findings are consistent with the idea that spatial representations form a generalized substrate for cognition: that space can act as a scaffold for learning abstract nonspatial concepts.Strengths:I enjoyed reading this manuscript, which is extremely well-written and well-presented. The writing is clear and concise throughout, and the figures do a great job of highlighting the key concepts. The issue of generalization is a core topic in neuroscience and psychology, relevant across a wide range of areas, and the findings will be of interest to researchers across areas in perception and cognitive science. It's also excellent to see that the hypotheses, methods, and analyses were pre-registered.The experiments that have been run are ingenious and thoughtful; I particularly liked the use of stimulus structures that allow for disentangling of one-dimensional and two-dimensional response patterns. The studies are also well-powered for detecting the effects of interest. The model-based statistical analyses are thorough and appropriate throughout (and it's good to see model recovery analysis too). The findings themselves are clear-cut: I have little doubt about the robustness and replicability of these data.Weaknesses:I have only one significant concern regarding this manuscript, which relates to the interpretation of the findings. The findings are taken to suggest that "space may serve as a 'scaffold', allowing people to visualize and manipulate nonspatial concepts" (p13). However, I think the data may be amenable to an alternative possibility. I wonder if it's possible that, for the visual and auditory stimuli, participants naturally tended to attend to one feature dimension and ignore the other - i.e., there may have been a (potentially idiosyncratic) difference in salience between the feature dimensions that led to participants learning the feature sequence in a one-dimensional way (akin to the 'overshadowing' effect in associative learning: e.g., see Mackintosh, 1976, "Overshadowing and stimulus intensity", Animal Learning and Behaviour). By contrast, we are very used to thinking about space as a multidimensional domain, in particular with regard to two-dimensional vertical and horizontal displacements. As a result, one would naturally expect to see more evidence of two-dimensional representation (allowing for rotational generalization) for spatial than nonspatial domains.In this view, the impact of spatial pre-training and (particularly) mapping is simply to highlight to participants that the auditory/visual stimuli comprise two separable (and independent) dimensions. Once they understand this, during subsequent training, they can learn about sequences on both dimensions, which will allow for a 2D representation and hence rotational generalization - as observed in Experiments 2 and 3. This account also anticipates that mapping alone (as in Experiment 4) could be sufficient to promote a 2D strategy for auditory and visual domains.This "attention to dimensions" account has some similarities to the "spatial scaffolding" idea put forward in the article, in arguing that experience of how auditory/visual feature manifolds can be translated into a spatial representation helps people to see those domains in a way that allows for rotational generalization. Where it differs is that it does not propose that space provides a *scaffold* for the development of the nonspatial representations, i.e., that people represent/learn the nonspatial information in a spatial format, and this is what allows them to manipulate nonspatial concepts. Instead, the "attention to dimensions" account anticipates that ANY manipulation that highlights to participants the separable-dimension nature of auditory/visual stimuli could facilitate 2D representation and hence rotational generalization. For example, explicit instruction on how the stimuli are constructed may be sufficient, or pre-training of some form with each dimension separately, before they are combined to form the 2D stimuli.I'd be interested to hear the authors' thoughts on this account - whether they see it as an alternative to their own interpretation, and whether it can be ruled out on the basis of their existing data.

We thank the Reviewer for their comments. We agree with the Reviewer that the “attention to dimensions” hypothesis is an interesting alternative explanation. However, we believe that the results of our control experiments Exp. 2c and Exp. 3c are incompatible with this alternative explanation.

In Exp. 2c, participants are pre-trained in the visual modality and then tested in the auditory modality. In the multimodal association task, participants have to associate the auditory stimuli and the visual stimuli: on each trial, they hear a sound and then have to click on the corresponding visual stimulus. It is thus necessary to pay attention to both auditory dimensions and both visual dimensions to perform the task. To give an example, the task might involve mapping the fundamental frequency and the amplitude modulation of the auditory stimulus to the colour and the shape of the visual stimulus, respectively. If participants pay attention to only one dimension, this would lead to a maximum of 25% accuracy on average (because they would be at chance on the other dimension, with four possible options). We observed that 30/50 participants reached an accuracy > 50% in the multimodal association task in Exp. 2c. This means that we know for sure that at least 60% of the participants paid attention to both dimensions of the stimuli. Nevertheless, there was a clear difference between participants that received a visual pre-training (Exp. 2c) and those who received a spatial pre-training (Exp. 2a) (frequency of 1D vs 2D models between conditions, BF > 100 in near transfer and far transfer). In fact, only 3/50 participants were best fit by a 2D model when vision was the pre-training modality compared to 29/50 when space was the pre-training modality. Thus, the benefit of the spatial pre-training cannot be due solely to a shift in attention toward both dimensions.

This effect was replicated in Exp. 3c. Similarly, 33/48 participants reached an accuracy > 50% in the multimodal association task in Exp. 3c, meaning that we know for sure that at least 68% of the participants actually paid attention to both dimensions of the stimuli. Again, there was a clear difference between participants who received a visual pre-training (frequency of 1D vs 2D models between conditions, Exp. 3c) and those who received a spatial pre-training (Exp. 3a) (BF > 100 in near transfer and far transfer).

Thus, we believe that the alternative explanation raised by the Reviewer is not supported by our data. We have added a paragraph in the discussion:

“One alternative explanation of this effect could be that the spatial pre-training encourages participants to attend to both dimensions of the non-spatial stimuli. By contrast, pretraining in the visual or auditory domains (where multiple dimensions of a stimulus may be relevant less often naturally) encourages them to attend to a single dimension. However, data from our control experiments Exp. 2c and Exp. 3c, are incompatible with this explanation. Around ~65% of the participants show a level of performance in the multimodal association task (>50%) which could only be achieved if they were attending to both dimensions (performance attending to a single dimension would yield 25% and chance performance is at 6.25%). This suggests that participants are attending to both dimensions even in the visual and auditory mapping case.”

**Reviewer #2 (Public Review):**
Summary:In this manuscript, L&S investigates the important general question of how humans achieve invariant behavior over stimuli belonging to one category given the widely varying input representation of those stimuli and more specifically, how they do that in arbitrary abstract domains. The authors start with the hypothesis that this is achieved by invariance transformations that observers use for interpreting different entries and furthermore, that these transformations in an arbitrary domain emerge with the help of the transformations (e.g. translation, rotation) within the spatial domain by using those as "scaffolding" during transformation learning. To provide the missing evidence for this hypothesis, L&S used behavioral category learning studies within and across the spatial, auditory, and visual domains, where rotated and translated 4-element token sequences had to be learned to categorize and then the learned transformation had to be applied in new feature dimensions within the given domain. Through single- and multiple-day supervised training and unsupervised tests, L&S demonstrated by standard computational analyses that in such setups, space and spatial transformations can, indeed, help with developing and using appropriate rotational mapping whereas the visual domain cannot fulfill such a scaffolding role.Strengths:The overall problem definition and the context of spatial mapping-driven solution to the problem is timely. The general design of testing the scaffolding effect across different domains is more advanced than any previous attempts clarifying the relevance of spatial coding to any other type of representational codes. Once the formulation of the general problem in a specific scientific framework is done, the following steps are clearly and logically defined and executed. The obtained results are well interpretable, and they could serve as a good stepping stone for deeper investigations. The analytical tools used for the interpretations are adequate. The paper is relatively clearly written.Weaknesses:Some additional effort to clarify the exact contribution of the paper, the link between analyses and the claims of the paper, and its link to previous proposals would be necessary to better assess the significance of the results and the true nature of the proposed mechanism of abstract generalization.(1) Insufficient conceptual setup: The original theoretical proposal (the Tolman-Eichenbaum-Machine, Whittington et al., Cell 2020) that L&S relate their work to proposes that just as in the case of memory for spatial navigation, humans and animals create their flexible relational memory system of any abstract representation by a conjunction code that combines on the one hand, sensory representation and on the other hand, a general structural representation or relational transformation. The TEM also suggests that the structural representation could contain any graph-interpretable spatial relations, albeit in their demonstration 2D neighbor relations were used. The goal of L&S's paper is to provide behavioral evidence for this suggestion by showing that humans use representational codes that are invariant to relational transformations of non-spatial abstract stimuli and moreover, that humans obtain these invariances by developing invariance transformers with the help of available spatial transformers. To obtain such evidence, L&S use the rotational transformation. However, the actual procedure they use actually solved an alternative task: instead of interrogating how humans develop generalizations in abstract spaces, they demonstrated that if one defines rotation in an abstract feature space embedded in a visual or auditory modality that is similar to the 2D space (i.e. has two independent dimensions that are clearly segregable and continuous), humans cannot learn to apply rotation of 4-piece temporal sequences in those spaces while they can do it in 2D space, and with co-associating a one-to-one mapping between locations in those feature spaces with locations in the 2D space an appropriate shaping mapping training will lead to the successful application of rotation in the given task (and in some other feature spaces in the given domain). While this is an interesting and challenging demonstration, it does not shed light on how humans learn and generalize, only that humans CAN do learning and generalization in this, highly constrained scenario. This result is a demonstration of how a stepwise learning regiment can make use of one structure for mapping a complex input into a desired output. The results neither clarify how generalizations would develop in abstract spaces nor the question of whether this generalization uses transformations developed in the abstract space. The specific training procedure ensures success in the presented experiments but the availability and feasibility of an equivalent procedure in a natural setting is a crucial part of validating the original claim and that has not been done in the paper.

We thank the Reviewer for their detailed comments on our manuscript. We reply to the three main points in turn.

First, concerning the conceptual grounding of our work, we would point out that the TEM model (Whittington et al., 2020), however interesting, is not our theoretical starting point. Rather, as we hope the text and references make clear, we ground our work in theoretical work from the 1990/2000s proposing that space acts as a scaffold for navigating abstract spaces (such as Gärdenfors, 2000). We acknowledge that the TEM model and other experimental work on the implication of the hippocampus, the entorhinal cortex and the parietal cortex in relational transformations of nonspatial stimuli provide evidence for this general theory. However, our work is designed to test a more basic question: whether there is behavioural evidence that space scaffolds learning in the first place. To achieve this, we perform behavioural experiments with causal manipulation (spatial pre-training vs no spatial pre-training) have the potential to provide such direct evidence. This is why we claim that:

“This theory is backed up by proof-of-concept computational simulations [13], and by findings that brain regions thought to be critical for spatial cognition in mammals (such as the hippocampal-entorhinal complex and parietal cortex) exhibit neural codes that are invariant to relational transformations of nonspatial stimuli. However, whilst promising, this theory lacks direct empirical evidence. Here, we set out to provide a strong test of the idea that learning about physical space scaffolds conceptual generalisation.“

Second, we agree with the Reviewer that we do not provide an explicit model for how generalisation occurs, and how precisely space acts as a scaffold for building representations and/or applying the relevant transformations to non-spatial stimuli to solve our task. Rather, we investigate in our Exp. 2-4 which aspects of the training are necessary for rotational generalisation to happen (and conclude that a simple training with the multimodal association task is sufficient for ~20% participants). We now acknowledge in the discussion the fact that we do not provide an explicit model and leave that for future work:

“We acknowledge that our study does not provide a mechanistic model of spatial scaffolding but rather delineate which aspects of the training are necessary for generalisation to happen.”

Finally, we also agree with the Reviewer that our task is non-naturalistic. As is common in experimental research, one must sacrifice the naturalistic elements of the task in exchange for the control and the absence of prior knowledge of the participants. We have decided to mitigate as possible the prior knowledge of the participants to make sure that our task involved learning a completely new task and that the pre-training was really causing the better learning/generalisation. The effects we report are consistent across the experiments so we feel confident about them but we agree with the Reviewer that an external validation with more naturalistic stimuli/tasks would be a nice addition to this work. We have included a sentence in the discussion:

“All the effects observed in our experiments were consistent across near transfer conditions (rotation of patterns within the same feature space), and far transfer conditions (rotation of patterns within a different feature space, where features are drawn from the same modality). This shows the generality of spatial training for conceptual generalisation. We did not test transfer across modalities nor transfer in a more natural setting; we leave this for future studies.”

(2) Missing controls: The asymptotic performance in experiment 1 after training in the three tasks was quite different in the three tasks (intercepts 2.9, 1.9, 1.6 for spatial, visual, and auditory, respectively; p. 5. para. 1, Fig 2BFJ). It seems that the statement "However, our main question was how participants would generalise learning to novel, rotated exemplars of the same concept." assumes that learning and generalization are independent. Wouldn't it be possible, though, that the level of generalization depends on the level of acquiring a good representation of the "concept" and after obtaining an adequate level of this knowledge, generalization would kick in without scaffolding? If so, a missing control is to equate the levels of asymptotic learning and see whether there is a significant difference in generalization. A related issue is that we have no information on what kind of learning in the three different domains was performed, albeit we probably suspect that in space the 2D representation was dominant while in the auditory and visual domains not so much. Thus, a second missing piece of evidence is the model-fitting results of the ⦰ condition that would show which way the original sequences were encoded (similar to Fig 2 CGK and DHL). If the reason for lower performance is not individual stimulus difficulty but the natural tendency to encode the given stimulus type by a combo of random + 1D strategy that would clarify that the result of the cross-training is, indeed, transferring the 2D-mapping strategy.

We agree with the Reviewer that a good further control is to equate performance during training. Thus, we have run a complementary analysis where we select only the participants that reach > 90% accuracy in the last block of training in order to equate asymptotic performance after training in Exp. 1. The results (see Author response image 1) replicates the results that we report in the main text: there is a large difference between groups (relative likelihood of 1D vs. 2D models, all BF > 100 in favour of a difference between the auditory and the spatial modalities, between the visual and the spatial modalities, in both near and far transfer, “decisive” evidence). We prefer not to include this figure in the paper for clarity, and because we believe this result is expected given the fact that 0/50 and 0/50 of the participants in the auditory and visual condition used a 2D strategy – thus, selecting subgroups of these participants cannot change our conclusions.

**Author response image 1. sa3fig1:** Results of Exp. 1 when selecting participants that reached > 90% accuracy in the last block of training. Captions are the same as Figure 2 of the main text.

Second, the Reviewer suggested that we run the model fitting analysis only on the ⦰ condition (training) in Exp. 1 to reveal whether participants use a 1D or a 2D strategy already during training. Unfortunately, we cannot provide the model fits only in the ⦰ condition in Exp. 1 because all models make the same predictions for this condition (see Fig S4). However, note that this is done by design: participants were free to apply whatever strategy they want during training; we then used the generalisation phase with the rotated stimuli precisely to reveal this strategy. Further, we do believe that the strategy used by the participants during training and the strategy during transfer are the same, partly because – starting from block #4 – participants have no idea whether the current trial is a training trial or a transfer trial, as both trial types are randomly interleaved with no cue signalling the trial type. We have made this clear in the methods:

“They subsequently performed 105 trials (with trialwise feedback) and 105 transfer trials including rotated and far transfer quadruplets (without trialwise feedback) which were presented in mixed blocks of 30 trials. Training and transfer trials were randomly interleaved, and no clue indicated whether participants were currently on a training trial or a transfer trial before feedback (or absence of feedback in case of a transfer trial).”

**Reviewer #3 (Public Review):**
Summary:Pesnot Lerousseau and Summerfield aimed to explore how humans generalize abstract patterns of sensory data (concepts), focusing on whether and how spatial representations may facilitate the generalization of abstract concepts (rotational invariance). Specifically, the authors investigated whether people can recognize rotated sequences of stimuli in both spatial and nonspatial domains and whether spatial pre-training and multi-modal mapping aid in this process.Strengths:The study innovatively examines a relatively underexplored but interesting area of cognitive science, the potential role of spatial scaffolding in generalizing sequences. The experimental design is clever and covers different modalities (auditory, visual, spatial), utilizing a two-dimensional feature manifold. The findings are backed by strong empirical data, good data analysis, and excellent transparency (including preregistration) adding weight to the proposition that spatial cognition can aid abstract concept generalization.Weaknesses:The examples used to motivate the study (such as "tree" = oak tree, family tree, taxonomic tree) may not effectively represent the phenomena being studied, possibly confusing linguistic labels with abstract concepts. This potential confusion may also extend to doubts about the real-life applicability of the generalizations observed in the study and raises questions about the nature of the underlying mechanism being proposed.

We thank the Reviewer for their comments. We agree that we could have explained ore clearly enough how these examples motivate our study. The similarity between “oak tree” and “family tree” is not just the verbal label. Rather, it is the arrangement of the parts (nodes and branches) in a nested hierarchy. Oak trees and family trees share the same relational structure. The reason that invariance is relevant here is that the similarity in relational structure is retained under rigid body transformations such as rotation or translation. For example, an upside-down tree can still be recognised as a tree, just as a family tree can be plotted with the oldest ancestors at either top or bottom. Similarly, in our study, the quadruplets are defined by the relations between stimuli: all quadruplets use the same basic stimuli, but the categories are defined by the relations between successive stimuli. In our task, generalising means recognising that relations between stimuli are the same despite changes in the surface properties (for example in far transfer). We have clarify that in the introduction:

“For example, the concept of a “tree” implies an entity whose structure is defined by a nested hierarchy, whether this is a physical object whose parts are arranged in space (such as an oak tree in a forest) or a more abstract data structure (such as a family tree or taxonomic tree). [...] Despite great changes in the surface properties of oak trees, family trees and taxonomic trees, humans perceive them as different instances of a more abstract concept defined by the same relational structure.”

Next, the study does not explore whether scaffolding effects could be observed with other well-learned domains, leaving open the question of whether spatial representations are uniquely effective or simply one instance of a familiar 2D space, again questioning the underlying mechanism.

We would like to mention that Reviewer #2 had a similar comment. We agree with both Reviewers that our task is non-naturalistic. As is common in experimental research, one must sacrifice the naturalistic elements of the task in exchange for the control and the absence of prior knowledge of the participants. We have decided to mitigate as possible the prior knowledge of the participants to make sure that our task involved learning a completely new task and that the pre-training was really causing the better learning/generalisation. The effects we report are consistent across the experiments so we feel confident about them but we agree with the Reviewer that an external validation with more naturalistic stimuli/tasks would be a nice addition to this work. We have included a sentence in the discussion:

“All the effects observed in our experiments were consistent across near transfer conditions (rotation of patterns within the same feature space), and far transfer conditions (rotation of patterns within a different feature space, where features are drawn from the same modality). This shows the generality of spatial training for conceptual generalisation. We did not test transfer across modalities nor transfer in a more natural setting; we leave this for future studies.”

Further doubt on the underlying mechanism is cast by the possibility that the observed correlation between mapping task performance and the adoption of a 2D strategy may reflect general cognitive engagement rather than the spatial nature of the task. Similarly, the surprising finding that a significant number of participants benefited from spatial scaffolding without seeing spatial modalities may further raise questions about the interpretation of the scaffolding effect, pointing towards potential alternative interpretations, such as shifts in attention during learning induced by pre-training without changing underlying abstract conceptual representations.

The Reviewer is concerned about the fact that the spatial pre-training could benefit the participants by increasing global cognitive engagement rather than providing a scaffold for learning invariances. It is correct that the participants in the control group in Exp. 2c have poorer performances on average than participants that benefit from the spatial pre-training in Exp. 2a and 2b. The better performances of the participants in Exp. 2a and 2b could be due to either the spatial nature of the pre-training (as we claim) or a difference in general cognitive engagement. .

However, if we look closely at the results of Exp. 3, we can see that the general cognitive engagement hypothesis is not well supported by the data. Indeed, the participants in the control condition (Exp. 3c) have relatively similar performances than the other groups during training. Rather, the difference is in the strategy they use, as revealed by the transfer condition. The majority of them are using a 1D strategy, contrary to the participants that benefited from a spatial pre-training (Exp 3a and 3b). We have included a sentence in the results:

“Further, the results show that participants who did not experience spatial pre-training were still engaged in the task, but were not using the same strategy as the participants who experienced spatial pre-training (1D rather than 2D). Thus, the benefit of the spatial pre-training is not simply to increase the cognitive engagement of the participants. Rather, spatial pre-training provides a scaffold to learn rotation-invariant representation of auditory and visual concepts even when rotation is never explicitly shown during pre-training.”

Finally, Reviewer #1 had a related concern about a potential alternative explanation that involved a shift in attention. We reproduce our response here: we agree with the Reviewer that the “attention to dimensions” hypothesis is an interesting (and potentially concerning) alternative explanation. However, we believe that the results of our control experiments Exp. 2c and Exp. 3c are not compatible with this alternative explanation.

Indeed, in Exp. 2c, participants are pre-trained in the visual modality and then tested in the auditory modality. In the multimodal association task, participants have to associate the auditory stimuli and the visual stimuli: on each trial, they hear a sound and then have to click on the corresponding visual stimulus. It is necessary to pay attention to both auditory dimensions and both visual dimensions to perform well in the task. To give an example, the task might involve mapping the fundamental frequency and the amplitude modulation of the auditory stimulus to the colour and the shape of the visual stimulus, respectively. If participants pay attention to only one dimension, this would lead to a maximum of 25% accuracy on average (because they would be at chance on the other dimension, with four possible options). We observed that 30/50 participants reached an accuracy > 50% in the multimodal association task in Exp. 2c. This means that we know for sure that at least 60% of the participants actually paid attention to both dimensions of the stimuli. Nevertheless, there was a clear difference between participants that received a visual pre-training (Exp. 2c) and those who received a spatial pre-training (Exp. 2a) (frequency of 1D vs 2D models between conditions, BF > 100 in near transfer and far transfer). In fact, only 3/50 participants were best fit by a 2D model when vision was the pre-training modality compared to 29/50 when space was the pre-training modality. Thus, the benefit of the spatial pre-training cannot be due solely to a shift in attention toward both dimensions.

This effect was replicated in Exp. 3c. Similarly, 33/48 participants reached an accuracy > 50% in the multimodal association task in Exp. 3c, meaning that we know for sure that at least 68% of the participants actually paid attention to both dimensions of the stimuli. Again, there was a clear difference between participants who received a visual pre-training (frequency of 1D vs 2D models between conditions, Exp. 3c) and those who received a spatial pre-training (Exp. 3a) (BF > 100 in near transfer and far transfer).

Thus, we believe that the alternative explanation raised by the Reviewer is not supported by our data. We have added a paragraph in the discussion:

“One alternative explanation of this effect could be that the spatial pre-training encourages participants to attend to both dimensions of the non-spatial stimuli. By contrast, pretraining in the visual or auditory domains (where multiple dimensions of a stimulus may be relevant less often naturally) encourages them to attend to a single dimension. However, data from our control experiments Exp. 2c and Exp. 3c, are incompatible with this explanation. Around ~65% of the participants show a level of performance in the multimodal association task (>50%) which could only be achieved if they were attending to both dimensions (performance attending to a single dimension would yield 25% and chance performance is at 6.25%). This suggests that participants are attending to both dimensions even in the visual and auditory mapping case.”

Conclusions:The authors successfully demonstrate that spatial training can enhance the ability to generalize in nonspatial domains, particularly in recognizing rotated sequences. The results for the most part support their conclusions, showing that spatial representations can act as a scaffold for learning more abstract conceptual invariances. However, the study leaves room for further investigation into whether the observed effects are unique to spatial cognition or could be replicated with other forms of well-established knowledge, as well as further clarifications of the underlying mechanisms.Impact:The study's findings are likely to have a valuable impact on cognitive science, particularly in understanding how abstract concepts are learned and generalized. The methods and data can be useful for further research, especially in exploring the relationship between spatial cognition and abstract conceptualization. The insights could also be valuable for AI research, particularly in improving models that involve abstract pattern recognition and conceptual generalization.In summary, the paper contributes valuable insights into the role of spatial cognition in learning abstract concepts, though it invites further research to explore the boundaries and specifics of this scaffolding effect.
**Reviewer #1 (Recommendations For The Authors):**
Minor issues / typos:P6: I think the example of the "signed" mapping here should be "e.g., ABAB maps to one category and BABA maps to another", rather than "ABBA maps to another" (since ABBA would always map to another category, whether the mapping is signed or unsigned).

Done.

P11: "Next, we asked whether pre-training and mapping were systematically associated with 2Dness...". I'd recommend changing to: "Next, we asked whether accuracy during pre-training and mapping were systematically associated with 2Dness...", just to clarify what the analyzed variables are.

Done.

P13, paragraph 1: "only if the features were themselves are physical spatial locations" either "were" or "are" should be removed.

Done.

P13, paragraph 1: should be "neural representations of space form a critical substrate" (not "for").

Done.

**Reviewer #2 (Recommendations For The Authors):**
The authors use in multiple places in the manuscript the phrases "learn invariances" (Abstract), "formation of invariances" (p. 2, para. 1), etc. It might be just me, but this feels a bit like 'sloppy' wording: we do not learn or form invariances, rather we learn or form representations or transformations by which we can perform tasks that require invariance over particular features or transformation of the input such as the case of object recognition and size- translation- or lighting-invariance. We do not form size invariance, we have representations of objects and/or size transformations allowing the recognition of objects of different sizes. The authors might change this way of referring to the phenomenon.

We respectfully disagree with this comment. An invariance occurs when neurons make the same response under different stimulation patterns. The objects or features to which a neuron responds is shaped by its inputs. Those inputs are in turn determined by experience-dependent plasticity. This process is often called “representation learning”. We think that our language here is consistent with this status quo view in the field.

**Reviewer #3 (Recommendations For The Authors):**
I understand that the objective of the present experiment is to study our ability to generalize abstract patterns of sensory data (concepts). In the introduction, the authors present examples like the concept of a "tree" (encompassing a family tree, an oak tree, and a taxonomic tree) and "ring" to illustrate the idea. However, I am sceptical as to whether these examples effectively represent the phenomena being studied. From my perspective, these different instances of "tree" do not seem to relate to the same abstract concept that is translated or rotated but rather appear to share only a linguistic label. For instance, the conceptual substance of a family tree is markedly different from that of an oak tree, lacking significant overlap in meaning or structure. Thus, to me, these examples do not demonstrate invariance to transformations such as rotations.To elaborate further, typically, generalization involves recognizing the same object or concept through transformations. In the case of abstract concepts, this would imply a shared abstract representation rather than a mere linguistic category. While I understand the objective of the experiments and acknowledge their potential significance, I find myself wondering about the real-world applicability and relevance of such generalizations in everyday cognitive functioning. This, in turn, casts some doubt on the broader relevance of the study's results. A more fitting example, or an explanation that addresses my concerns about the suitability of the current examples, would be beneficial to further clarify the study's intent and scope.

Response in the public review.

Relatedly, the manuscript could benefit from greater clarity in defining key concepts and elucidating the proposed mechanism behind the observed effects. Is it plausible that the changes observed are primarily due to shifts in attention induced by the spatial pre-training, rather than a change in the process of learning abstract conceptual invariances (i.e., modifications to the abstract representations themselves)? While the authors conclude that spatial pre-training acts as a scaffold for enhancing the learning of conceptual invariances, it raises the question: does this imply participants simply became more focused on spatial relationships during learning, or might this shift in attention represent a distinct strategy, and an alternative explanation? A more precise definition of these concepts and a clearer explanation of the authors' perspective on the mechanism underlying these effects would reduce any ambiguity in this regard.

Response in the public review.

I am wondering whether the effectiveness of spatial representations in generalizing abstract concepts stems from their special nature or simply because they are a familiar 2D space for participants. It is well-established that memory benefits from linking items to familiar locations, a technique used in memory training (method of loci). This raises the question: Are we observing a similar effect here, where spatial dimensions are the only tested familiar 2D spaces, while the other 2 spaces are simply unfamiliar, as also suggested by the lower performance during training (Fig.2)? Would the results be replicable with another well-learned, robustly encoded domain, such as auditory dimensions for professional musicians, or is there something inherently unique about spatial representations that aids in bootstrapping abstract representations?On the other side of the same coin, are spatial representations qualitatively different, or simply more efficient because they are learned more quickly and readily? This leads to the consideration that if visual pre-training and visual-to-auditory mapping were continued until a similar proficiency level as in spatial training is achieved, we might observe comparable performance in aiding generalization. Thus, the conclusion that spatial representations are a special scaffold for abstract concepts may not be exclusively due to their inherent spatial nature, but rather to the general characteristic of well-established representations. This hypothesis could be further explored by either identifying alternative 2D representations that are equally well-learned or by extending training in visual or auditory representations before proceeding with the mapping task. At the very least I believe this potential explanation should be explored in the discussion section.

Response in the public review.

I had some difficulty in following an important section of the introduction: "... whether participants can learn rotationally invariant concepts in nonspatial domains, i.e., those that are defined by sequences of visual and auditory features (rather than by locations in physical space, defined in Cartesian or polar coordinates) is not known." This was initially puzzling to me as the paragraph preceding it mentions: "There is already good evidence that nonspatial concepts are represented in a translation invariant format." While I now understand that the essential distinction here is between translation and rotation, this was not immediately apparent upon first reading. This crucial distinction, especially in the context of conceptual spaces, was not clearly established before this point in the manuscript. For better clarity, it would be beneficial to explicitly contrast and define translation versus rotation in this particular section and stress that the present study concerns rotations in abstract spaces.

Done.

The multi-modal association is crucial for the study, however to my knowledge, it is not depicted or well explained in the main text or figures (Results section). In my opinion, the details of this task should be explained and illustrated before the details of the associated results are discussed.

We have included an illustration of a multimodal association trial in Fig. S3B.

**Author response image 2. sa3fig2:** 

The observed correlation between the mapping task performance and the adoption of a 2D strategy is logical. However, this correlation might not exclusively indicate the proposed underlying mechanism of spatial scaffolding. Could it also be reflective of more general factors like overall performance, attention levels, or the effort exerted by participants? This alternative explanation suggests that the correlation might arise from broader cognitive engagement rather than specifically from the spatial nature of the task. Addressing this possibility could strengthen the argument for the unique role of spatial representations in learning abstract concepts, or at least this alternative interpretation should be mentioned.

Response in the public review.

To me, the finding that ~30% of participants benefited from the spatial scaffolding effect for example in the auditory condition merely through exposure to the mapping (Fig 4D), without needing to see the quadruplets in the spatial modality, was somewhat surprising. This is particularly noteworthy considering that only ~60% of participants adopted the 2D strategy with exposure to rotated contingencies in Experiment 3 (Fig 3D). How do the authors interpret this outcome? It would be interesting to understand their perspective on why such a significant effect emerged from mere exposure to the mapping task.I appreciate the clarity Fig.1 provides in explaining a challenging experimental setup. Is it possible to provide example trials, including an illustration that shows which rotations produce the trail and an intuitive explanation that response maps onto the 1D vs 2D strategies respectively, to aid the reader in better understanding this core manipulation?I like that the authors provide transparency by depicting individual subject's data points in their results figures (e.g. Figs. 2 B, F, J). However, with an n=~50 per condition, it becomes difficult to intuit the distribution, especially for conditions with higher variance (e.g., Auditory). The figures might be more easily interpretable with alternative methods of displaying variances, such as violin plots per data point, conventional error shading using 95%CIs, etc.Why are the authors not reporting exact BFs in the results sections at least for the most important contrasts?While I understand why the authors report the frequencies for the best model fits, this may become difficult to interpret in some sections, given the large number of reported values. Alternatives or additional summary statistics supporting inference could be beneficial.

As the Reviewer states, there are a large number of figures that we can report in this study. We have chosen to keep this number at a minimum to be as clear as possible. To illustrate the distribution of individual data points, we have opted to display only the group's mean and standard error (the standard errors are included, but the substantial number of participants per condition provides precise estimates, resulting in error bars that can be smaller than the mean point). This decision stems from our concern that including additional details could lead to a cluttered representation with unnecessary complexity. Finally, we report what we believe to be the critical BFs for the comprehension of the reader in the main text, and choose a cutoff of 100 when BFs are high (corresponding to the label “decisive” evidence, some BFs are larger than 1012). All the exact BFs are in the supplementary for the interested readers.